# Private Geometric Median

**Mahdi Haghifam**[*]        **Thomas Steinke**[†]        **Jonathan Ullman**[‡]

## Abstract

In this paper, we study differentially private (DP) algorithms for computing the geometric median (GM) of a dataset: Given $n$ points, $x_1, \ldots, x_n$ in $\mathbb{R}^d$, the goal is to find a point $\theta$ that minimizes the sum of the Euclidean distances to these points, i.e., $\sum_{i=1}^{n} \|\theta - x_i\|_2$. Off-the-shelf methods, such as DP-GD, require strong a priori knowledge locating the data within a ball of radius $R$, and the excess risk of the algorithm depends linearly on $R$. In this paper, we ask: can we design an efficient and private algorithm with an excess error guarantee that scales with the (unknown) radius containing the majority of the datapoints? Our main contribution is a pair of polynomial-time DP algorithms for the task of private GM with an excess error guarantee that scales with the effective diameter of the datapoints. Additionally, we propose an inefficient algorithm based on the inverse smooth sensitivity mechanism, which satisfies the more restrictive notion of pure DP. We complement our results with a lower bound and demonstrate the optimality of our polynomial-time algorithms in terms of sample complexity.

## 1 Introduction

Differentially private (DP) convex optimization is a fundamental task where we approximately minimize a data-dependent convex loss function while limiting what can be learned about individual data points. The predominant algorithm for DP convex optimization is DP (stochastic) gradient descent, or DP-(S)GD, for short [SCS13; BST14]. Given a dataset $\mathbf{X}^{(n)}$ which contains private information, and a loss function $F(\theta; \mathbf{X}^{(n)})$, DP-(S)GD starts with an initial value $\theta_0 \in \mathbb{R}^d$ and iteratively updates it using $\theta_{t+1} = \Pi_\Theta \big( \theta_t - \eta \cdot \big( \nabla_{\theta_t} F(\theta_t; \mathbf{X}^{(n)}) + \xi_t \big) \big)$ where $\eta > 0$ is the step size, $\xi_t$ is noise to ensure DP, $\Theta \subseteq \mathbb{R}^d$ is a closed convex feasible set, and $\Pi_\Theta$ is the Euclidean projection operator. In the most general setting of Lipschitz convex functions, the excess error depends *linearly* on the radius of the set $\Theta$, and this linear dependence is necessary in the worst-case [BST14]. This linear dependence is problematic because we can think of the diameter of the set $\Theta$ as capturing a measure of the uncertainty we have about the location of the minimizer, and we want our algorithm to perform well even with a high degree of uncertainty. This linear dependence can be improved under certain unrealistically strong assumptions, such as strong convexity, but it is unclear whether we can improve the dependence on the radius under weaker, more natural conditions. In this paper, as a step towards answering this question, we identify a simple, optimization task—computing the *geometric median*—where we can exponentially improve the dependence on the radius.

We study private algorithms for computing the *geometric median (GM)* of a dataset: We are given a set of $n$ data points $\mathbf{X}^{(n)} \triangleq (x_1, \ldots, x_n) \in (\mathbb{R}^d)^n$, where $x_i$ represents the private information of

---

[*]Khoury College of Computer Sciences, Northeastern University. Supported by a Khoury College of Computer Sciences Distinguished Postdoctoral Fellowship. `m.haghifam@northeastern.edu`

[†]Google DeepMind.

[‡]Khoury College of Computer Sciences, Northeastern University. Supported by NSF awards CNS-2232692 and CNS-2247484.

| Algorithm | Privacy | Utility $[F(\mathcal{A}_n(\mathbf{X}^{(n)}); \mathbf{X}^{(n)})]$ | Run-time | Samples |
|---|---|---|---|---|
| LocDPSGD (Section 3.1) | Approx | $\left(1 + \frac{\sqrt{d}}{n\varepsilon}\right)\mathsf{OPT}$ | $\boxed{n^2 \log(R/r)} + n^2 d$ | $\frac{\sqrt{d\log(R/r)}}{\varepsilon}$ |
| LocDPCuttingPlane (Section 3.2) | Approx | $\left(1 + \frac{\sqrt{d}}{n\varepsilon}\right)\mathsf{OPT}$ | $\boxed{n^2 \log(R/r)} + nd^2 + d^{2+\omega}$ | $\frac{\sqrt{d\log(R/r)}}{\varepsilon}$ |
| SInvS (Section 4) | Pure | $\left(1 + \frac{d\log(R/r)}{n\varepsilon}\right)\mathsf{OPT}$ | Exponential | $\frac{d\log(R/r)}{\varepsilon}$ |
| Baseline: DP-(S)GD | Approx | $\mathsf{OPT} + \frac{R\sqrt{d}}{\varepsilon}$ | $n^2 d$ | N/A |

**Table 1:** Summary of our results. Here $\mathsf{OPT} = \arg\min_{\theta \in \mathbb{R}^d} F(\theta; \mathbf{X}^{(n)})$ denotes the optimal loss and $\omega$ is the matrix-multiplication exponent. The highlighted part is the runtime of the warm-up phase which is the same for LocDPSGD and LocDPCuttingPlane. We also assume that $\max_{i \in [n]} |\mathbf{X}^{(n)} \cap \mathcal{B}_d(x_i, r)| < 3n/4$. (See Section 3.1, Section 3.2, and Section 4 for the general results without this restriction.) For readability, we omit logarithmic factors that depend on $n$ and $d$.

one individual, and we are interested in approximately solving the following optimization problem:

$$\theta^\star \triangleq \mathsf{GM}(\mathbf{X}^{(n)}) \in \arg\min_{\theta \in \mathbb{R}^d} F(\theta; \mathbf{X}^{(n)}), \quad \text{where,} \quad F(\theta; \mathbf{X}^{(n)}) \triangleq \sum_{i \in [n]} \|\theta - x_i\|_2. \tag{1}$$

The geometric median generalizes the standard one-dimensional median. The geometric median is a useful tool for robust estimation and aggregation, because it is less sensitive to outliers than the mean of the data, i.e., it is a nontrivial estimator even when $\leq 49\%$ of the input data is arbitrarily corrupted. These properties make GM a popular tool for designing robust versions of distributed optimization methods [CSX17; WLCG20; FGGPS22; AHJSDT22; PKH22; EFGH23], boosting the confidence of weakly concentrated estimators [Min15], clustering [BMM03], etc.

**Baseline for Private GM.** Since the geometric median is the minimizer of a Lipschitz convex loss function, we can privately approximate it using the standard approach of DP-(S)GD. In particular, if we know a priori that all the data points lie in a known ball of radius $R$ (without loss of generality this ball is centered at the origin, i.e., $\|x_i\|_2 \leq R$ for every $i \in [n]$), then DP-(S)GD guarantees $(\varepsilon, \delta)$-DP with the following excess error [BST14]:

$$F(\mathrm{DPGD}_n(\mathbf{X}^{(n)}); \mathbf{X}^{(n)}) - F(\theta^\star; \mathbf{X}^{(n)}) = O\left(\frac{R\sqrt{d\log(1/\delta)}}{\varepsilon}\right). \tag{2}$$

As discussed in the beginning of this section, this guarantee has a significant drawback: the excess error of the algorithm depends *linearly* on the radius $R$ of the a priori bound on the data. This bound could be very loose; it does not scale with the data. Can we do better? What quantity should the excess error guarantee scale with?

It is known that the GM is inside the convex hull of the datapoints. However, this convex hull can have a very large diameter due to a small number of *outliers*, while *most* of the datapoints live in a ball with a small diameter. A key property of GM is robustness to outliers, so we want our accuracy guarantee to also be robust to some outliers. Specifically, if $\geq 51\%$ of the points lie in a ball of diameter $\Delta \ll R$ then the geometric median is $O(\Delta)$ far from that ball (see Lemma C.6 for a more precise statement). Thus, we aim to design a DP algorithm whose error is proportional to the actual scale of the majority of the data, rather than the a priori worst-case bound. However, the algorithm designer does not have a priori knowledge of the location or diameter of a ball that contains most of the data; the algorithm must discover this information from the data. This prompts the following question: *Can we design an efficient and private algorithm with an excess error guarantee that scales with the radius that contains majority of the datapoints?* Our results provide a positive answer.

## 1.1 Contributions

Our main contribution is a pair of *polynomial-time* DP algorithms for approximating the geometric median with an excess error guarantee that scales with the effective diameter of the datapoints. Also, the sample complexity and the runtime of our algorithms depend logarithmically on the a priori bound $R$. Both of our algorithms achieve the same excess error bounds up to logarithmic factors, but have incomparable running times. We also give a simple numerical experiment on synthetic data as a proof of concept that our algorithm improves over DP-(S)GD, as presdicted by the theory. In

terms of optimality, we show the our proposed algorithms is optimal in terms of sample complexity. Furthermore, we propose an algorithm based on the *inverse smooth sensitivity* mechanism for the private geometric median problem that satisfies the more restrictive notion of *pure* DP. Below, we give an overview of these algorithms and the techniques involved.

**Polynomial-Time Algorithms.** Both of our algorithms for the private geometric median problem are two-phase algorithms: in the first phase, which we refer to as *warm-up*, the algorithm shrinks the feasible set to a ball whose diameter is proportional to what we call the *quantile radius* in time that depends logarithmically on $R$. The second phase, which we call *fine-tuning*, uses the output of the warm-up algorithm to further improve the error.

First, we formalize the notion of the quantile radius as the radius of the smallest ball containing sufficiently many points.

**Definition 1.1** (Quantile Radius). Fix a dataset $\mathbf{X}^{(n)} = (x_1, \ldots, x_n) \in (\mathbb{R}^d)^n$ and $\theta \in \mathbb{R}^d$. For every $\gamma \in [0, 1]$, define $\Delta_{\gamma n}(\theta) \triangleq \min\{\Delta : |i \in [n] : \|x_i - \theta\| \leq \Delta| \geq \gamma n\}$.

To motivate the idea behind our algorithms, assume the algorithm designer *knew* a ball, with center $\theta_0$ and radius $\hat{\Delta}$ such that $\|\theta_0 - \theta^\star\| \leq O(\hat{\Delta})$ and $\hat{\Delta} = \widetilde{O}(\Delta_{4n/5}(\theta^\star))$. Then, running DP-(S)GD over this ball would give excess error $O(\Delta_{4n/5}(\theta^\star)\sqrt{d}/\varepsilon)$. This guarantee is particularly interesting as the excess error scales with the quantile radius and not the largest possible norm of any point. Also, by definition of the quantile radius and the geometric median loss function, we have that $F(\theta^\star; \mathbf{X}^{(n)}) \geq (1 - \gamma)n\Delta_{\gamma n}(\theta^\star)$. This inequality shows that an algorithm whose excess error depends on $\Delta_{\gamma n}(\theta^\star)$ has a *multiplicative guarantee* rather than the standard additive guarantee for DP-(S)GD. This type of guarantee is particularly desirable for the geometric median since an algorithm with a multiplicative guarantee will be scale free and be adaptive to the niceness of the dataset. However, since we do not know such a pair $\theta_0$ and $\hat{\Delta}$ a priori, the objective of the warm-up algorithm is to privately find these quantities.

The warm-up algorithm is based on the following structural result: given a point $\theta$ that satisfies $\|\theta - \theta^\star\| \gtrsim \Delta_{3n/4}(\theta^\star)$, we have $F(\theta; \mathbf{X}^{(n)}) - F(\theta^\star; \mathbf{X}^{(n)}) \gtrsim \|\theta - \theta^\star\|$. (See Lemma 2.6 for a formal statement.) This result implies that, even though $F(\theta; \mathbf{X}^{(n)})$ is not a strongly convex function, we have a *growth condition* such that the excess error increases with the distance to the global minimizer, at least when the excess error is large enough. (In contrast, strong convexity would imply quadratic growth $F(\theta; \mathbf{X}^{(n)}) - F(\theta^\star; \mathbf{X}^{(n)}) \gtrsim \|\theta - \theta^\star\|^2$, rather than linear growth.) Intuitively, this growth condition allows us to take larger step sizes and make progress faster, consuming less of the privacy budget. However, since this growth condition only holds for $\theta$ that is more than $\Delta_{3n/4}(\theta^\star)$ away from the minimizer, which is a data-dependent property, we first need to develop a private algorithm to estimate $\Delta_{3n/4}(\theta^\star)$ in order to make use of this property. In Section 2.1, we develop an efficient algorithm, `RadiusFinder`, for this task, which is inspired by [NSV16]. Our procedure assumes that we know some potentially very small lower bound $r \leq \Delta_{3n/4}(\theta^\star)$, which is necessary by the impossibility results in [BNSV15]. Since the sample complexity of this procedure depends only on $\log(1/r)$, we can choose this parameter to be very small. In Section 2.1, we show how to eliminate this assumption at the cost of a small additive error. With high probability, `RadiusFinder` (see Theorem 2.4) outputs $\hat{\Delta}$ such that $\Delta_{3n/4}(\theta^\star) \leq \hat{\Delta} \leq O(\Delta_{4n/5}(\theta^\star))$. Having obtained $\hat{\Delta}$, the second step of the warm-up algorithm is finding a good initialization point. In Section 2.2, we propose `Localization`, based on DP-GD with *geometrically decaying step sizes*, to perform this task. Due to the growth condition we show that DP-GD makes a fast progress towards some point that is within $O(\Delta_{4n/5}(\theta^\star))$ from the optimizer: in $\log(R)$ iterations, with high probability, it outputs $\theta_0$ such that $\theta^\star$ is in the ball of radius $O(\hat{\Delta}) = O(\Delta_{4n/5}(\theta^\star))$ centered at $\theta_0$.

**DP Cutting Plane Method for Private GM.** The main drawback of using DP-SGD for the fine-tuning stage is that its run-time can be large when $n \gg d$. To address this, we design the second fine-tuning algorithm, `LocDPCuttingPlane`, based on private variant of the cutting plane method that has faster running time when $n$ is large. There are two challenges in the analysis: by using the noisy gradients, we cannot argue that the optimal point always lives in the intersection of the cutting planes, which is a crucial part of the standard analysis. The second challenge is that the cutting plane method is not a *descent* method in the sense that the loss function is not decreasing with the iteration, and we need to privately select an iterate with small loss. The challenge for developing the private variant here is that the loss $F(\theta; \mathbf{X}^{(n)})$ has sensitivity proportional to $R$, so running the exponential

mechanism in the natural way incurs loss proportional to $R$. We address both of these challenges and develop an algorithm whose excess error is proportional to $\Delta_{4n/5}(\theta^\star)$.

**Pure DP algorithm for Private Geometric Median Problem.** In Section 4, we propose a pure $(\varepsilon, 0)$-DP algorithm for the geometric median problem, albeit a computationally inefficient one. Our algorithm is based on the *inverse smooth sensitivity* mechanism of [AD20]. At a high level, the algorithm outputs $\theta \in \mathbb{R}^d$ with a probability proportional to $\exp(-\varepsilon \cdot \mathrm{len}(\mathbf{X}^{(n)}, \theta)/2)$ where $\mathrm{len}(\mathbf{X}^{(n)}, \theta)$ is the minimum number of data points from $\mathbf{X}^{(n)}$ that needs to be modified to obtain a dataset $\tilde{\mathbf{X}}^{(n)}$ such that the geometric median of $\tilde{\mathbf{X}}^{(n)}$ be equal $\theta$. Our analysis shows that the proposed mechanism outputs $\hat{\theta} = \mathrm{GM}(\tilde{\mathbf{X}}^{(n)})$ such that $\tilde{\mathbf{X}}^{(n)}$ and $\mathbf{X}^{(n)}$ differ in at most $k^\star = O(d \log(R)/\varepsilon)$ with a high probability. Then, by a careful sensitivity analysis, we show $\|\hat{\theta} - \theta^\star\|$ can be upper bounded by the $F(\theta^\star; \mathbf{X}^{(n)})$. Using this result we provide an algorithm with a multiplicative guarantee. Moreover, we show $\|\hat{\theta} - \theta^\star\|$ is upper bounded $O(\Delta_{\gamma n}(\theta^\star))$ for some $\gamma \in (1/2, 1]$.

**Lower bound on the Sample Complexity.** We show every $(\varepsilon, \delta)$-DP algorithm requires $\tilde{\Omega}(\sqrt{d}/\varepsilon)$ samples so that it satisfies $\mathbb{E}_{\hat{\theta} \sim \mathcal{A}_n(\mathbf{X}^{(n)})}[F(\hat{\theta}; \mathbf{X}^{(n)})] \leq (1+\alpha) \min_{\theta \in \mathbb{R}^d} F(\theta, \mathbf{X}^{(n)})$ for a constant $\alpha$. This result shows that the sample complexity of our polynomial-time algorithms is nearly optimal.

A summary of the results is provided in Table 1, comparing the proposed algorithms in terms of privacy, utility, runtime, and sample complexity. As discussed earlier, algorithms with error adaptive to the quantile radius can achieve a nearly multiplicative guarantee. The utility column in Table 1 compares the algorithms based on the achievable $\alpha_{\mathrm{mul}}$ and $\alpha_{\mathrm{add}}$ in order to $F(\mathcal{A}_n(\mathbf{X}^{(n)}); \mathbf{X}^{(n)}) \leq (1+\alpha_{\mathrm{mul}})F(\theta^\star; \mathbf{X}^{(n)}) + \alpha_{\mathrm{add}}$ with a high probability.

## 1.2 Related Work

DP convex optimization is a well-studied problem [CMS11; KST12; BST14; ACGMMTZ16; STU17; FKT20]. There has been significant interest in developing new algorithms that offer improved guarantees compared to DP-(S)GD for specific problem classes or by leveraging additional information. For instance, [LUZ20; SSTT21; ABGMU22; BMS22] demonstrate that for linear models the dependency of the excess error on the dimension can be improved, [GHST24; ABL23] study the impact of the second-order information on the convergence, [KDRT21; AGMRSSSTT22; GHNOSTTW23] explore the impact of public data, etc. The current paper addresses a drawback of DP-(S)GD, namely, the linear dependence of the excess error on the distance from the initializer to the optimal point in non-strongly convex settings.

Another related line of work to our warm-up strategy is private averaging of [NSV16; NS18; CKMST21; TCKMS22]. The advantage of the algorithm proposed in this work is its simplicity while being optimal in terms of sample complexity: we exploit a structural property of the geometric median and show that running DPGD with the geometrically decaying stepsizes can yield a suitable initialization point without the need for preprocessing steps such as filtering [CKMST21; TCKMS22], coordinate-wise discretization [NSV16], hashing [NS18], etc. The proposed quantile radius can be seen as a robust notion of radius proposed in [BHI02].

In one dimension (i.e., $d = 1$), private versions of the median are well studied [DNPR10; BNS13; BNSV15; DNRR15; BDRS18; ALMM19; KLMNS20; ASSU23; CLNSS23]. In particular, these works improve the dependence on the a priori bound $R$ to $\log^* R$, rather than $\log R$ in our results.

## 1.3 Notation

Let $d \in \mathbb{N}$. For a vector $x \in \mathbb{R}^d$, $\|x\|$ denotes the $\ell_2$ norm of $x$. We use the following notation for the ball of radius $R$: $\mathcal{B}_d(a, R) = \{x \in \mathbb{R}^d : \|x - a\| \leq R\}$. Also, $\mathcal{B}_d^\infty(a, R)$ denotes $\{x \in \mathbb{R}^d : \|x - a\|_\infty \leq R\}$. We refer to $\mathcal{B}_d(0, R) = \mathcal{B}_d(R)$, similarly, it holds for $\mathcal{B}_d^\infty(0, R) = \mathcal{B}_d^\infty(R)$. Let $\langle \cdot, \cdot \rangle$ denote the standard inner product in $\mathbb{R}^d$. For a convex and closed subset $\Theta \subseteq \mathbb{R}^d$, let $\Pi_\Theta : \mathbb{R}^d \to \Theta$ be the Euclidean projection operator, given by $\Pi_\Theta(x) = \arg\min_{y \in \Theta} \|y - x\|_2$. For a (measurable) space $\mathcal{R}$, $\mathcal{M}_1(\mathcal{R})$ denotes the set of all probability measures on $\mathcal{R}$. Let $\mathcal{Z}$ be the data space and let $\Theta \subseteq \mathbb{R}^d$ be the parameter space. Let $f : \Theta \times \mathcal{Z} \to \mathbb{R}$ be a loss function. We say $f$ is *L-Lipschitz* iff there exists $L \in \mathbb{R}$ such that $\forall z \in \mathcal{Z}, \forall w, v \in \Theta : |f(w, z) - f(v, z)| \leq L \|w - v\|$.

### 1.4 Notions of DP

**Definition 1.2.** Let $\varepsilon > 0$ and $\delta \in [0, 1)$. A randomized mechanism $\mathcal{A}_n : \mathcal{Z}^n \to \mathcal{M}_1(\Theta)$ is $(\varepsilon, \delta)$-DP, iff, for every neighbouring dataset (i.e., replacement) $\mathbf{X} \in \mathcal{Z}^n$ and $\mathbf{X}' \in \mathcal{Z}^n$, and for every measurable subset $M \subseteq \Theta$, it holds $\mathbb{P}_{\theta \sim \mathcal{A}_n(\mathbf{X})}(\theta \in M) \leq e^\varepsilon \cdot \mathbb{P}_{\theta \sim \mathcal{A}_n(\mathbf{X}')}(\theta \in M) + \delta$.

For some of our privacy analysis, we use concentrated differential privacy [DR16; BS16], as it provides a simpler composition theorem – the privacy parameter $\rho$ adds up when we compose.

**Definition 1.3** ([BS16, Def. 1.1]). A randomized mechanism $\mathcal{A} : \mathcal{Z}^n \to \mathcal{M}_1(\mathcal{R})$ is $\rho$-zCDP, iff, for every neighbouring dataset (i.e., replacement) $\mathbf{X} \in \mathcal{Z}^n$ and $\mathbf{X}' \in \mathcal{Z}^n$, and for every $\alpha \in (1, \infty)$, it holds $D_\alpha(\mathcal{A}_n(\mathbf{X}) \| \mathcal{A}_n(\mathbf{X}')) \leq \rho\alpha$, where $D_\alpha(\mathcal{A}_n(\mathbf{X}) \| \mathcal{A}_n(\mathbf{X}'))$ is the $\alpha$-Renyi divergence between $\mathcal{A}_n(\mathbf{X})$ and $\mathcal{A}_n(\mathbf{X}')$.

We should think of $\rho \approx \varepsilon^2$: to attain $(\varepsilon, \delta)$-DP, it suffices to set $\rho = \frac{\varepsilon^2}{4 \log(1/\delta) + 4\varepsilon}$ [BS16, Lem. 3.5].

**Lemma 1.4** ([BS16, Prop. 1.3]). *Assume we have a randomized mechanism* $\mathcal{A} : \mathcal{Z} \to \mathcal{M}_1(\mathcal{R})$ *that satisfies* $\rho$-*zCDP, then for every* $\delta > 0$, $\mathcal{A}$ *is* $(\rho + 2\sqrt{\rho \log(1/\delta)}, \delta)$-*DP.*

## 2 Private Localization

In this section, we present the proposed algorithm for the warm-up stage; it has two steps: *Private Estimation of Quantile Radius* and *Private Localization*.

### 2.1 Step 1: Private Estimation of Quantile Radius

Algorithm 1 describes our private algorithm `RadiusFinder` for quantile radius estimation.

---

**Algorithm 1** $\texttt{RadiusFinder}_n$

---

1: Inputs: data set $\mathbf{X}^{(n)} \in (\mathcal{B}_d(R))^n$, fraction $\gamma \in (1/2, 1]$, privacy budget $\rho$-zCDP, failure probability $\beta$, discretization error $0 < r < R$ .
2: $m = \lceil \gamma n \rceil$.
3: For every $\nu \geq 0$ and $i \in [n]$, let

$$N_i(\nu) \triangleq |\mathbf{X}^{(n)} \cap \mathcal{B}_d(x_i, \nu)|.$$

$\qquad \qquad \qquad \qquad \qquad \qquad \qquad \triangleright$ Number of datapoints within distance of $\nu$ from $x_i$

4: For every $\nu \geq 0$, define

$$N(\nu) \triangleq \frac{1}{m} \max_{\text{distinct}\{i_1,\ldots,i_m\} \subseteq [n]} \{N_{i_1}(\nu) + \cdots + N_{i_m}(\nu)\}.$$

5: Grid $= \{r, 2r, 4r \cdots, 2^{\lceil \log(\frac{2R}{r}) \rceil} r\}$.
6: Queries $= \{N(v) : v \in \text{Grid}\}$
7:
$$\hat{i} = \texttt{AboveThreshold}\left(\text{Queries}, \rho, m + \frac{18}{\sqrt{2\rho}} \log\left(\frac{2}{\beta} \cdot \left\lceil \log\left(\frac{2R}{r}\right) \right\rceil\right)\right)$$

$\qquad \qquad \qquad \qquad \qquad \qquad \qquad \qquad \qquad \qquad \qquad \qquad \triangleright$ Algorithm 7

8: Output $\hat{\Delta} = 2^{\hat{i}} r$ if $\hat{i} \neq \texttt{Fail}$; else Output $\texttt{Fail}$.

---

*Remark* 2.1. The runtime of `RadiusFinder` is $\Theta((n^2 + n \log(n)) \log(\lceil R/r \rceil))$: First, we need to compute the pairwise distances which take $n^2$ time. Then, for a fixed $\nu$, we can compute $N(\nu)$ using the pairwise distances in time $\Theta(n^2)$. To compute $N(\nu)$, we need to sort $\{N_i(\nu)\}_{i \in [n]}$, in $\Theta(n \log(n))$ time, and pick top $m$. Finally, we need to repeat this for each $\nu \in [r, \ldots, 2^{\lceil \log(\frac{2R}{r}) \rceil} r]$. $\qquad \triangleleft$

Notice that Algorithm 1 uses the datapoints as centers for computing the number of the datapoints in a given distance. The privacy proof of Algorithm 1 is based on the following lemma.

**Lemma 2.2.** *Fix* $n \in \mathbb{N}$. *For every dataset* $\mathbf{X}^{(n)}$, *for every* $1/2 \leq \gamma \leq 1$ *and for every fixed* $\nu$, *the query* $N(\nu) \triangleq \frac{1}{m} \max_{\{i_1,\ldots,i_m\} \subseteq [n]} \{N_{i_1}(\nu) + \cdots + N_{i_m}(\nu)\}$, *has a sensitivity upper-bounded by* 3 *where* $m = \lceil \gamma n \rceil$ *and* $N_i(\nu) \triangleq |\mathbf{X}^{(n)} \cap \mathcal{B}_d(x_i, \nu)|$. *Here* $\mathcal{B}_d(x, \nu) := \{y \in \mathbb{R}^d : \|y - x\| \leq \nu\}$.

The objective of Algorithm 1 is to privately approximate $\Delta_{\gamma n}(\theta^\star)$. Nonetheless, Algorithm 1 relies on computing the pairwise distances between datapoints. The following lemma elucidates why computing these pairwise distances serves as an effective proxy for computing $\Delta_{\gamma n}(\theta^\star)$.

**Lemma 2.3.** *Fix $n \in \mathbb{N}$, $1 \le m \le n$, $\gamma_1, \gamma_2 \in (1/2, 1]$ such that $\gamma_2 \ge \gamma_1$, and dataset $\mathbf{X}^{(n)}$. For every $\nu \ge 0$, define $N(\nu) \triangleq \frac{1}{m} \max_{\{i_1,\dots,i_m\} \subseteq [n]} \{N_{i_1}(\nu) + \cdots + N_{i_m}(\nu)\}$, where $N_i(\nu) \triangleq |\mathbf{X}^{(n)} \cap \mathcal{B}_d(x_i, \nu)|$. Let $\theta^\star = \mathrm{GM}(\mathbf{X}^{(n)})$. For every $\hat{\nu}$ such that $N(\hat{\nu}) \ge \lceil \gamma_1 n \rceil$ and $N(\hat{\nu}/2) < \lceil \gamma_2 n \rceil$, we have*

$$\Delta_{\gamma_1 n}(\theta^\star) \cdot \frac{2\gamma_1 - 1}{4\gamma_1 - 1} \le \hat{\nu} \le 4\Delta_{\gamma_2 n}(\theta^\star).$$

Using these two lemmas, in the next theorem, we present the privacy and utility guarantees of Algorithm 1. As we are interested in finding the smallest radius, we use the standard `AboveThreshold` from [DNRRV09; DR+14] as a subroutine in Algorithm 1. The algorithmic description of `AboveThreshold` is provided in Appendix B for completeness.

**Theorem 2.4.** *Let `RadiusFinder`$_n$ denote Algorithm 1. Fix $d \in \mathbb{N}$, $R > 0$, $r > 0$, $\beta \in (0, 1]$, and $\rho > 0$. Then, for every $n \in \mathbb{N}$ and every dataset $\mathbf{X}^{(n)} \in (\mathcal{B}_d(R))^n$ the output of `RadiusFinder`$_n$ satisfies $\rho$-zCDP. Also, the output of `RadiusFinder`$_n$ satisfies the following utility guarantees:*

1. *Given $n > \frac{18}{(1-\gamma)\sqrt{2\rho}} \log(4/\beta)$, then $\mathbb{P}\left(\Delta_{\gamma n}(\theta^\star)\frac{2\gamma-1}{4\gamma-1} \le \hat{\Delta}\right) \ge 1 - \beta$.*

2. *Assume that the data points satisfies $N(r) < m$. Let $\tilde{\gamma} \triangleq \min\{\gamma + \frac{1}{n}\frac{36}{\sqrt{2\rho}}\log(2(\lceil\log(\frac{2R}{r})\rceil + 1)/\beta), 1\}$, then, given $n > \frac{18}{(1-\gamma)\sqrt{2\rho}} \log(4/\beta)$, we have*

$$\mathbb{P}\left(\Delta_{\gamma n}(\theta^\star)\frac{2\gamma-1}{4\gamma-1} \le \hat{\Delta} \le 4\Delta_{\tilde{\gamma}n}(\theta^\star)\right) \ge 1 - \frac{5}{2}\beta.$$

3. *Let $\tilde{\gamma} \triangleq \min\{\gamma + \frac{1}{n}\frac{36}{\sqrt{2\rho}}\log(2(\lceil\log(\frac{2R}{r})\rceil + 1)/\beta), 1\}$. Given $n > \frac{18}{(1-\gamma)\sqrt{2\rho}} \log(4/\beta)$, we have*

$$\mathbb{P}\left(\Delta_{\gamma n}(\theta^\star)\frac{2\gamma-1}{4\gamma-1} \le \hat{\Delta} \text{ and } \left\{\hat{\Delta} \le 4\Delta_{\tilde{\gamma}n}(\theta^\star) \text{ or } \hat{\Delta} = r\right\}\right) \ge 1 - 2\beta.$$

*Remark* 2.5. A sufficient condition for $N(r) < m$ in Item 2 is that $\max_{i \in [n]} |\mathbf{X}^{(n)} \cap \mathcal{B}_d(x_i, r)| < m = \lceil \gamma n \rceil$. Intuitively, this means that no data point should have a significant portion of other data points within a ball of radius $r$ centered on it. ◁

## 2.2 Step 2: Fast Localization

In the second step of the warm-up phase, we develop a fast algorithm for finding a good initialization point using the private estimate of the quantile radius. The main structural result that we use for the algorithm design is stated in the next lemma.

**Lemma 2.6.** *Fix $n \in \mathbb{N}$, $\mathbf{X}^{(n)} \in (\mathbb{R}^d)^n$ and $\theta_1, \theta_0 \in \mathbb{R}^d$. For every $\gamma \in [0, 1]$, define $\Delta_{\gamma n}(\theta_0) \triangleq \min\{r \ge 0 : |i \in [n] : \|x_i - \theta_0\| \le r| \ge \gamma n\}$. Assume there exists $\zeta \ge 0$ such that $F(\theta_1; \mathbf{X}^{(n)}) - F(\theta_0; \mathbf{X}^{(n)}) \le \zeta n$. Then, for every $\gamma \in (1/2, 1]$, we have*

$$(2\gamma - 1)\|\theta_1 - \theta_0\| - 2\gamma\Delta_{\gamma n}(\theta_0) \le \zeta$$

To gain some intuition behind Lemma 2.6, let us instantiate $\theta_0 = \theta^\star$. This result implies that for a $\theta \in \mathbb{R}^d$ such that $\|\theta - \theta^\star\| \gtrsim \Delta_{\gamma n}(\theta^\star)$, the loss function of the geometric median satisfies $F(\theta; \mathbf{X}^{(n)}) - F(\theta^\star; \mathbf{X}^{(n)}) \gtrsim \|\theta - \theta^\star\|$. Using this result, we propose Algorithm 2 for finding a good initialization. The next theorem states the privacy and utility guarantees of Algorithm 2.

**Theorem 2.7.** *Let `Localization`$_n$ denote Algorithm 2. Fix $d \in \mathbb{N}$, $R > 0$, $r > 0$, $\rho > 0$, and $\beta \in (0, 1)$. Then for every dataset $\mathbf{X}^{(n)} \in (\mathcal{B}_d(R))^n$ the outputs of `Localization`$_n$ satisfies $\rho$-zCDP. Moreover, let $(\hat{\theta}, \hat{\Delta}) = $ `Localization`$_n(\mathbf{X}^{(n)}, \rho, r, \beta)$ and define random set $\Theta_{loc} = \{\theta \in \mathcal{B}_d(R) : \|\theta - \hat{\theta}\| \le 25\hat{\Delta}\}$. Then, given*

$$n \ge \Omega\left(\max\left\{\frac{\sqrt{d\log(\lceil R/r \rceil)}}{\sqrt{\rho}}\sqrt{\log\left(\frac{\log(\lceil R/r \rceil)}{\beta}\right)}, \frac{1}{\sqrt{\rho}}\log\left(\frac{\lceil R/r \rceil}{\beta}\right)\right\}\right),$$

---

**Algorithm 2** $\texttt{Localization}_n$

---

1: Inputs: dataset $\mathbf{X}^{(n)} \in (\mathcal{B}_d(R))^n$, privacy parameters $\rho$-zCDP, discretization error $r$, failure
    probability $\beta$

2: $\gamma = 3/4$

3: $\hat{\Delta} = \texttt{RadiusFinder}_n\left(\mathbf{X}^{(n)}, \gamma, \frac{\rho}{2}, \frac{\beta}{2}, r\right)$                          $\triangleright$ Algorithm 1

4: **if** $\hat{\Delta} = \texttt{Fail}$ **then**

5:     Output $\texttt{Fail}$ and Halt.

6: $k_{\text{wu}} = \frac{1}{\log(2)} \log\left(R/\hat{\Delta}\right)$                $\triangleright$ $k_{\text{wu}} \leq \frac{1}{\log(2)} \log(R/r)$ with probability one

7: $\theta_0 = 0 \in \mathbb{R}^d$, $T_{\text{wu}} = 500$, $\text{rad}_0 = R$

8: **for** $t \in \{0, \ldots, k_{\text{wu}} - 1\}$ **do**

9:     $\Theta_t = \{\theta \in \mathcal{B}_d(R) : \|\theta - \theta_t\| \leq \text{rad}_t\}$

10:    $\eta_t = \text{rad}_t \sqrt{\frac{2dk_{\text{wu}}}{3\rho n^2}}$

11:    $\theta_{t+1} = \texttt{DPGD}\left(\theta_t, \mathbf{X}^{(n)}, \frac{\rho}{2k_{\text{wu}}}, \Theta_t, \eta_t, T_{\text{wu}}\right)$            $\triangleright$ Algorithm 6

12:    $\text{rad}_{t+1} = \frac{1}{2}\text{rad}_t + 12\hat{\Delta}$

13: Output $\theta_{k_{\text{wu}}}$ and $\hat{\Delta}$.

---

we have $\mathbb{P}\left(\theta^\star \in \Theta_{loc} \text{ and } \Delta_{0.75n}(\theta^\star) \leq 4\hat{\Delta} \text{ and } \left\{\hat{\Delta} \leq 4\Delta_{0.8n}(\theta^\star) \text{ or } \hat{\Delta} = r\right\}\right) \geq 1 - 2\beta$. Also, assuming that the datapoints satisfies $\max_{i \in [n]} |\mathbf{X}^{(n)} \cap \mathcal{B}_d(x_i, r)| < 3n/4$, we have

$$\mathbb{P}\left(\theta^\star \in \Theta_{loc} \text{ and } \Delta_{0.75n}(\theta^\star) \leq 4\hat{\Delta} \text{ and } \hat{\Delta} \leq 4\Delta_{0.8n}(\theta^\star)\right) \geq 1 - 2\beta.$$

## 3 Private Fine-tuning

In Section 2, we developed an algorithm for the warm-up stage. The output of the warm-up stage is $\theta_0$ and radius $\hat{\Delta}$ such that $\|\theta_0 - \theta^\star\| \leq O(\hat{\Delta})$ and $\hat{\Delta} = \widetilde{O}(\Delta_{4n/5}(\theta^\star))$ as formalized in Theorem 2.7. In this section, we build upon the output of the warm-up algorithm to develop two polynomial-time algorithms for the fine-tuning stage.

### 3.1 Fine-tuning Using DPGD

Our first algorithm is based on DP-GD [BST14]. The main ideas behind Algorithm 3 is as follows: 1) from the utility guarantee of the warm-up phase in Theorem 2.7, the distance of the initialization and $\theta^\star$ only depends on $\hat{\Delta}$, i.e., it does not depend on $R$, 2) By definition of the quantile radius in Definition 1.1 and Equation (1), we have that $F(\theta^\star; \mathbf{X}^{(n)}) \geq (1 - \gamma)n\Delta_{\gamma n}(\theta^\star)$, 3) in the case that the data satisfies some regularity conditions, we have $\hat{\Delta} \leq 4\Delta_{0.8n}(\theta^\star)$ from Theorem 2.7. The next theorem summarizes the utility and privacy guarantees of this algorithm.

---

**Algorithm 3** $\texttt{LocDPGD}_n$

---

1: Inputs: dataset $\mathbf{X}^{(n)} \in (\mathcal{B}_d(R))^n$, privacy parameters $\rho$-zCDP, discretization error $r$, failure probability $\beta$.

2: $\theta_0, \hat{\Delta} = \texttt{Localization}_n\left(\mathbf{X}^{(n)}, \frac{\rho}{2}, r, \frac{\beta}{2}\right)$               $\triangleright$ Algorithm 2

3: $\Theta_0 = \{\theta \in \mathcal{B}_d(R) : \|\theta - \theta_0\| \leq 25\hat{\Delta}\}$

4: $\eta_{\text{ft}} = 50\hat{\Delta}\sqrt{\frac{d}{6\rho n^2}}$ and $T_{\text{ft}} = \frac{n^2\rho}{256d}$

5: $\hat{\theta} = \texttt{DPGD}\left(\theta_0, \mathbf{X}^{(n)}, \frac{\rho}{2}, \Theta_0, \eta_{\text{ft}}, T_{\text{ft}}\right)$                 $\triangleright$ Algorithm 6

6: Output $\hat{\theta}$

---

**Theorem 3.1.** *Let* $\texttt{LocalizedDPGD}_n$ *denote Algorithm 3. For every* $d \in \mathbb{N}$, $R > 0$, $r > 0$, $\rho > 0$, *and* $\beta \in (0, 1]$, $\mathcal{A} = \{\texttt{LocalizedDPGD}_n\}_{n \geq 1}$ *satisfies the following: for every* $n \in \mathbb{N}$ *and every*

*dataset* $\mathbf{X}^{(n)} \in (\mathcal{B}_d(R))^n$ *the output of* $\texttt{LocalizedDPGD}_n$ *satisfies* $\rho$-*zCDP. Also, given*

$$n \geq \Omega\left(\max\left\{\frac{\sqrt{d\log(\lceil R/r\rceil)}}{\sqrt{\rho}}\sqrt{\log\left(\frac{\log(\lceil R/r\rceil)}{\beta}\right)}, \frac{1}{\sqrt{\rho}}\log\left(\frac{\lceil R/r\rceil}{\beta}\right)\right\}\right),$$

*we have*

$$\mathbb{P}\left(F\left(\hat{\theta}; \mathbf{X}^{(n)}\right) \leq \left(1 + O\left(\frac{\sqrt{d}}{n\sqrt{\rho}}\sqrt{\log(1/\beta)}\right)\right)F\left(\theta^\star; \mathbf{X}^{(n)}\right) + O\left(\sqrt{\frac{d\log(1/\beta)}{\rho}}\right)r\right) \geq 1 - 2\beta.$$

*Moreover, given that the datapoints satisfies* $\max_{i\in[n]}|\mathbf{X}^{(n)} \cap \mathcal{B}_d(x_i, r)| < 3n/4$, *we have*

$$\mathbb{P}\left(F\left(\hat{\theta}; \mathbf{X}^{(n)}\right) \leq \left(1 + O\left(\frac{\sqrt{d}}{n\sqrt{\rho}}\sqrt{\log(1/\beta)}\right)\right)F\left(\theta^\star; \mathbf{X}^{(n)}\right)\right) \geq 1 - 2\beta,$$

*where* $\hat{\theta}$ *is the output of Algorithm 3.*

### 3.2 Fine-tuning Using Noisy Cutting Plane Method

In this section, we present the second fine-tuning algorithm: $\texttt{LocDPCuttingPlane}$ of Algorithm 4. This algorithm is based on the well-known cutting plane method [New65; Lev65; Nes98].

---

**Algorithm 4** $\texttt{LocDPCuttingPlane}_n$

---

1: Inputs: dataset $\mathbf{X}^{(n)} \in (\mathcal{B}_d(R))^n$, privacy parameters $(\varepsilon, \delta)$-DP, discretization error $r$, failure probability $\beta$
2: $\rho = \frac{\varepsilon^2}{16\log(2/\delta)+8\varepsilon}$
3: $\theta_0, \hat{\Delta} = \texttt{Localization}_n\left(\mathbf{X}^{(n)}, \frac{\rho}{2}, r, \min\{\frac{\beta}{3}, \frac{\delta}{2}\}\right)$ $\qquad\qquad\qquad\qquad\qquad$ ▷ Algorithm 2
4: $\Theta_0 = \{\theta \in \mathcal{B}_d(R) : \|\theta - \theta_0\| \leq 25\hat{\Delta}\}$
5: $k_{\text{ft}} = \Theta\left(\frac{d}{\tau}\log\left(\frac{n\sqrt{\tau\cdot\rho}}{\sqrt{d}} + \sqrt{d}\right)\right)$ $\qquad\qquad\qquad\qquad$ ▷ See Assumption 1 for definition of $\tau$
6: **for** $t \in \{0, \ldots, k_{\text{ft}} - 1\}$ **do**
7: $\quad$ $\theta_t = \texttt{Centre}(\Theta_t)$ $\qquad\qquad\qquad\qquad\qquad\qquad\qquad\qquad\qquad$ ▷ See Assumption 1
8: $\quad$ $\xi_{\text{dir},t} \sim \mathcal{N}\left(0, \frac{k_{\text{ft}}}{\rho}\mathbb{I}_d\right)$
9: $\quad$ $\Theta_{t+1} = \left\{\theta \in \Theta_t \big| \left\langle \nabla F(\theta_t; \mathbf{X}^{(n)}) + \xi_{\text{dir},t}, \theta - \theta_t \right\rangle < 0\right\}$
10: Define Probability Measure: $\pi(t) \propto \exp\left(-\frac{\varepsilon}{448\hat{\Delta}}F\left(\theta_t; \mathbf{X}^{(n)}\right)\right)$ $\quad$ for $t \in \{0, \ldots, k_{\text{ft}} - 1\}$
11: Output $\theta_{\hat{t}}$ where $\hat{t} \sim \pi$

---

Similar to non-private cutting plane method, $\texttt{LocDPCuttingPlane}$ is not a descent algorithm. As a result, we need to devise a mechanism for selecting an iterate with minimal loss. In the next lemma, we provide a bespoke analysis of the exponential mechanism with the score function $F(\theta; \mathbf{X}^{(n)})$ defined in Equation (1). Note that the sensitivity of $F(\theta; \mathbf{X}^{(n)})$ is $R$. However, the next result demonstrates that through a novel analysis of the sensitivity of $F(\theta; \mathbf{X}^{(n)})$, the noise scale due to privacy can be significantly reduced. Proof can be found in Appendix E.

**Lemma 3.2.** *Let* $\varepsilon \in \mathbb{R}$, $k \in \mathbb{N}$, *and* $d \in \mathbb{N}$ *be constants. Let* $\Theta \subseteq \mathbb{R}^d$ *be a set with a bounded diameter of* **diam**. *Let* $\mathbf{X}^{(n)} \in (\mathbb{R}^d)^n$ *be a dataset and* $\theta^\star \in \text{GM}(\mathbf{X}^{(n)})$. *Let* $\{\theta_1, \ldots, \theta_k\} \subseteq \Theta$ *be* $k$ *fixed vectors. Also, assume that* $\theta^\star \in \Theta$. *Let* $\Delta$ *be such that* $3\Delta_{3n/4}(\theta^\star) + 2$**diam** $\leq \Delta$. *Consider the following probability measure over* $\{1, \ldots, k\}$:

$$\pi(i; \mathbf{X}^{(n)}) = \frac{\exp\left(-\frac{\varepsilon}{2\Delta}F(\theta_i; \mathbf{X}^{(n)})\right)}{\sum_{j\in[k]}\exp\left(-\frac{\varepsilon}{2\Delta}F(\theta_j; \mathbf{X}^{(n)})\right)}, \quad i \in [k].$$

*1. Let* $\hat{i} \sim \pi(\cdot; \mathbf{X}^{(n)})$ *and* $OPT \triangleq \min_{i\in[k]}\{F(\theta_i; \mathbf{X}^{(n)}) - F(\theta^\star; \mathbf{X}^{(n)})\}$. *Then, for every* $\beta \in (0, 1]$, *we have*

$$\mathbb{P}\left(F(\theta_{\hat{i}}; \mathbf{X}^{(n)}) - F(\theta^\star; \mathbf{X}^{(n)}) \leq OPT + \frac{2\Delta}{\varepsilon}\log(k/\beta)\right) \geq 1 - \beta.$$

2. Let $\tilde{\mathbf{X}}^{(n)}$ be a dataset of size $n$ that differs in one sample from $\mathbf{X}^{(n)}$. Then, for every $i \in [k]$, we have

$$\exp(-\varepsilon)\pi(i; \tilde{\mathbf{X}}^{(n)}) \leq \pi(i; \mathbf{X}^{(n)}) \leq \exp(\varepsilon)\pi(i; \tilde{\mathbf{X}}^{(n)}).$$

The next theorem provides the privacy guarantee of Algorithm 4. The privacy analysis differs from the rest of the algorithms in the paper. This deviation arises from the fact that for analyzing the privacy guarantee of Line 10 of Algorithm 4, we use Lemma 3.2. Notice that the guarantee in Lemma 3.2 holds provided that $\Theta_0$, defined in Line 4 of Algorithm 4, satisfies $\theta^\star \in \Theta_0$. Ergo, the privacy guarantee of Algorithm 4 only satisfies *approximate-DP*.

**Theorem 3.3.** *Let* $\texttt{LocDPCuttingPlane}_n$ *denote Algorithm 4. Fix* $d \in \mathbb{N}$, $R > 0$, $r > 0$, $\varepsilon > 0$, $\delta \in (0, 1]$, *and* $\beta \in (0, 1]$. *Then, for every* $n \in \mathbb{N}$ *and every dataset* $\mathbf{X}^{(n)} \in (\mathcal{B}_d(R))^n$ *the output of* $\texttt{LocDPCuttingPlane}_n$ *satisfies* $(\varepsilon, \delta)$-*DP.*

We also make the following assumption about the performance of $\texttt{Centre}$ subroutine in Algorithm 4.

**Assumption 1.** *There exists some* $\tau \in (0, 1]$ *such that for all* $t \in \{0, \dots, k_{\mathit{ft}} - 1\}$, *the subroutine of* $\texttt{Centre}$ *in Algorithm 4 satisfies* $vol(\Theta_{t+1}) \leq (1 - \tau)vol(\Theta_t)$. *Furthermore, the time for calling the routine* $\texttt{Centre}$ *is* $T_c$.

Using the John Ellipsoid [Joh14] as the $\texttt{Centre}$ makes $\tau$ a dimension independent constant and $T_c = \tilde{O}(d^{1+\omega})$ (by [LSW15]). Now we are ready to state the utility guarantee of Algorithm 4.

**Theorem 3.4.** *Let* $\texttt{LocDPCuttingPlane}_n$ *denote Algorithm 4. For every* $d \in \mathbb{N}$, $R > 0$, $r > 0$, $\varepsilon > 0$, $\delta \in (0, 1]$, *and* $\beta \in (0, 1]$, $\mathcal{A} = \{\texttt{LocDPCuttingPlane}_n\}_{n \geq 1}$ *satisfies the following: for every* $n \in \mathbb{N}$ *and every dataset* $\mathbf{X}^{(n)} \in (\mathcal{B}_d(R))^n$, *given*

$$n \geq \Omega\left(\max\left\{\frac{\sqrt{d\log(\lceil R/r \rceil)}}{\sqrt{\rho}}\sqrt{\log\left(\frac{\log(\lceil R/r \rceil)}{\beta}\right)}, \frac{1}{\sqrt{\rho}}\log\left(\frac{\lceil R/r \rceil}{\beta}\right)\right\}\right),$$

*where* $\rho = \frac{\varepsilon^2}{16\log(2/\delta)+8\varepsilon}$, *we have the following: Let* $\kappa \triangleq \frac{n\sqrt{\rho}}{\sqrt{d}} + \sqrt{d}$ *and* $\alpha = O\left(\sqrt{\frac{d\log(\kappa)}{\tau\rho}} \cdot \log\left(\frac{d\log(\kappa)}{\tau\beta}\right)\right)$. *Then,*

$$\mathbb{P}\left(F\left(\hat{\theta}; \mathbf{X}^{(n)}\right) \leq \left(1 + \frac{\alpha}{n}\right)F(\theta^\star; \mathbf{X}^{(n)}) + r\alpha\right) \geq 1 - 3\beta,$$

*Moreover, assuming that the datapoints satisfies* $\max_{i \in [n]}|\mathbf{X}^{(n)} \cap \mathcal{B}_d(x_i, r)| < 3n/4$, *we have*

$$\mathbb{P}\left(F\left(\hat{\theta}; \mathbf{X}^{(n)}\right) \leq \left(1 + \frac{\alpha}{n}\right)F(\theta^\star; \mathbf{X}^{(n)})\right) \geq 1 - 3\beta,$$

*where* $\hat{\theta}$ *is the output of Algorithm 4.*

# 4 Pure-DP Algorithm for Geometric Median

In this section, we propose an algorithm based on the assumption that we have an access to an oracle that outputs an *exact* $\mathrm{GM}(\mathbf{X}^{(n)})$. Before presenting the algorithm, we need a definition: For two sequences of $\boldsymbol{a} = (a_1, \dots, a_n) \in (\mathbb{R}^d)^n$ and $\boldsymbol{b} = (b_1, \dots, b_n) \in (\mathbb{R}^d)^n$, we define the hamming distance as $\mathrm{d_H}(\boldsymbol{a}, \boldsymbol{b}) = \sum_{i=1}^n \mathbb{1}[a_i \neq b_i]$. The proposed algorithm is shown in Algorithm 5, and its utility and privacy guarantees are presented in the following theorem.

**Theorem 4.1.** *Let* $\texttt{SInvS}_n$ *denote the algorithm in Algorithm 5. Fix* $d \in \mathbb{N}$, $R > 0$, $r > 0$, *and* $\varepsilon > 0$. *Then, for every* $n \in \mathbb{N}$ *and every dataset* $\mathbf{X}^{(n)} \in (\mathcal{B}_d(R))^n$ *the output of* $\texttt{SInvS}_n$ *satisfies* $\varepsilon$-*DP. Also, for every* $\beta \in (0, 1)$ *and for every* $n > 2k^\star \triangleq 2\left\lfloor\frac{2}{\varepsilon}(\log(1/\beta) + d\log(R/r))\right\rfloor$, *with probability at least* $1 - \beta$, *we have:*

*1. The value of the cost function satisfies*

$$F(\hat{\theta}; \mathbf{X}^{(n)}) \leq \left(1 + \frac{4k^\star}{n - 2k^\star}\right)F(\theta^\star; \mathbf{X}^{(n)}) + nr.$$

---

**Algorithm 5** $\text{SInvS}_n$

---

1: Input: dataset $\mathbf{X}^{(n)} \in (\mathcal{B}_d(R))^n$, privacy parameter $\varepsilon$-DP, discretization error $r$.
2: For every $y \in \mathcal{B}_d(R)$

$$\text{len}_r(\mathbf{X}, y) \triangleq \min_{\tilde{\mathbf{X}} \in (\mathbb{R}^d)^n} \{d_{\text{H}}(\mathbf{X}^{(n)}, \tilde{\mathbf{X}}^{(n)}) \text{ such that } \exists z \in \mathcal{B}_d(y, r) \text{ with } \text{GM}(\tilde{\mathbf{X}}^{(n)}) = z\}$$

3: Define density: $d\pi(y) = \dfrac{\exp\left(-\frac{\varepsilon}{2} \cdot \text{len}_r(\mathbf{X}, y)\right)}{\int_{y \in \mathcal{B}_d(R)} \exp\left(-\frac{\varepsilon}{2} \cdot \text{len}_r(\mathbf{X}, y)\right) dy} \mathbb{1}[y \in \mathcal{B}_d(R)]$

4: Output $\hat{\theta} \sim \pi$

---

2. *In terms of distance,*

$$\left\| \hat{\theta} - \theta^\star \right\| \leq r + \min_{\gamma \in (1/2,1]:\gamma > \frac{k^\star}{n} + \frac{1}{2}} \frac{\Delta_{\gamma n}(\theta^\star)}{\sqrt{2(\gamma - k^\star/n) - (\gamma - k^\star/n)^2}}.$$

The proof of Theorem 4.1 is provided in Appendix F. The proof is based on showing that the output $\hat{\theta} = \text{GM}\left(\tilde{\mathbf{X}}^{(n)}\right)$ is such that $\tilde{\mathbf{X}}^{(n)}$ and $\mathbf{X}^{(n)}$ differ in at most $k^\star = O(d \log(R)/\varepsilon)$ datapoints with a high probability. Then, we use the properties of the geometric median to show that the sensitivity of GM to changing $k < n/2$ points can be bounded by the value of the optimal loss at $\theta^\star = \text{GM}(\mathbf{X}^{(n)})$.

**Lemma 4.2.** *For every $n \in \mathbb{N}$ and for every $k < \frac{n}{2}$, and for every $(x_1, \ldots, x_n, y_1, \ldots, y_k) \in (\mathbb{R}^d)^{n+k}$, define $\theta_0 = \text{GM}((x_1, \ldots, x_n))$ and $\theta_k = \text{GM}((x_1, \ldots, x_{n-k}, y_1, \ldots, y_k))$. Then,*
$$\|\theta_k - \theta_0\| \leq \frac{2}{n - 2k} F(\theta_0; (x_1, \ldots, x_n)).$$

## 5 Lower Bound on the Sample Complexity

In this section we prove a lower bound on the sample complexity of any $(\varepsilon, \delta)$-DP algorithm for the task of private geometric median with a multiplicative error.

**Theorem 5.1.** *Let $\varepsilon_0, \alpha_0, d_0$ be universal constants. Then, for every $\varepsilon \leq \varepsilon_0$, $\alpha \leq \alpha_0$, and $d \geq d_0$ and every $(\varepsilon, \delta)$-DP algorithm $\mathcal{A}_n : (\mathbb{R}^d)^n \to \mathcal{M}_1(\mathbb{R}^d)$ (with $\delta = \tilde{O}(\sqrt{d}/n)$) such that for every dataset $\mathbf{X}^{(n)} \in (\mathbb{R}^d)^n$ its output satisfies $\mathbb{E}_{\hat{\theta} \sim \mathcal{A}_n(\mathbf{X}^{(n)})}\left[F\left(\hat{\theta}; \mathbf{X}^{(n)}\right)\right] \leq (1 + \alpha) \min_{\theta \in \mathcal{B}_d^\infty(1)} F(\theta; \mathbf{X}^{(n)})$, we require $n = \tilde{\Omega}\left(\frac{\sqrt{d}}{\varepsilon}\right)$.*

This result, whose proof can be found in Appendix G, shows that the sample complexity of the proposed polynomial time algorithms is tight in terms of the dependence on $\varepsilon$ and $d$.

## 6 Numerical Example

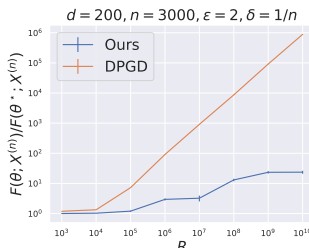

In this section, we numerically compare $\text{LocDPGD}_n$ (Algorithm 3) and DPGD on a synthetic dataset. The dataset consists of two subsets: one tightly clustered at a random location on $\mathcal{B}_d(R)$, and the other uniformly distributed over $\mathcal{B}_d(R)$. We plot $F(\theta; \mathbf{X}^{(n)})/F(\theta^\star; \mathbf{X}^{(n)})$ for both algorithms as $R$ varies. The results show that $\text{LocDPGD}_n$'s performance degrades more gracefully than DP-GD with increasing $R$. See Appendix H for experimental details and more results.

## 7 Conclusion and Limitations

In this paper, we presented three private algorithms for the geometric median task, ensuring an excess error guarantee that scales with the effective data scale. Our results open up many directions: we believe our warm-up algorithm has broader applications, and finding other problems where it can be used as a subroutine is interesting. Another direction is to characterize the optimal run-time: is it possible to develop a linear time algorithm, i.e. $\tilde{\Theta}(nd)$, with an optimal excess error?

## Acknowledgments

The authors would like to thank Jad Silbak, Eliad Tsfadia, and Mohammad Yaghini for helpful discussions.

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

# A Preliminaries

## A.1 Gradient of the Geometric Loss

$$\nabla_\theta(\|\theta - x\|) = \begin{cases} \frac{\theta - x}{\|\theta - x\|} & \theta \neq x \\ 0 & \theta = x \end{cases}. \tag{3}$$

## A.2 DP Gradient Descent (DPGD)

In this section, we provide the algorithmic description of DPGD and its privacy and utility analysis for completeness.

---

**Algorithm 6** DPGD

---

1: Inputs: initialization point $\theta_1 \in \mathbb{R}^d$, dataset $\mathbf{X}^{(n)} \in (\mathbb{R}^d)^{(n)}$, privacy budget $\rho$, feasible set $\Theta$, stepsize $\eta$, number of iterations $T$.
2: $\sigma^2 = \frac{T}{2\rho n^2}$
3: **for** $t \in \{1, \ldots, T\}$ **do**

$$\theta_{t+1} = \Pi_\Theta(\theta_t - \eta(\nabla F(\theta_t; \mathbf{X}^{(n)}) + \xi_t)),$$

where $\xi_t \sim \mathcal{N}(0, \sigma^2 I_d)$.
4: Output $\frac{1}{T}\sum_{t=1}^{T}\theta_t$

---

**Lemma A.1.** *Let $\Theta \subseteq \mathbb{R}^d$ be a closed and convex set with a finite diameter* diam. *Let $\ell : \Theta \times \mathcal{Z} \to \mathbb{R}$ be a loss function such that for every $z \in \mathcal{Z}$, $\ell(\cdot, z)$ is convex and L-Lipschitz. Let $\mathbf{X}^{(n)} = (z_1, \ldots, z_n) \in \mathcal{Z}^n$ and $\hat{L}_n(\theta) = \frac{1}{n}\sum_{i\in[n]}\ell(\theta, z_i)$. Consider DP-Gradient descent algorithm $\theta_{t+1} = \Pi_\Theta(\theta_t - \eta(\nabla\hat{L}_n(\theta_t) + \xi_t))$, where $\xi_t \sim \mathcal{N}(0, \sigma^2 I_d)$. Then, for every $T \in \mathbb{N}$, by setting $\eta = \mathsf{diam}\sqrt{\frac{d}{12L^2\rho n^2}}$, and $\sigma^2 = \frac{L^2 T}{2\rho n^2}$, we have the following: $\{\theta_t\}_{t\in[T]}$ satisfies $\rho$-zCDP. Also, for every $\beta > 0$, given $Td \geq \log(4/\beta)$ and $1 \leq \sqrt{56\log(2/\beta)}$, with probability at least $1 - \beta$, we have*

$$\hat{L}_n\left(\frac{1}{T}\sum_{t\in[T]}\theta_t\right) - \min_{\theta\in\Theta}\hat{L}_n(\theta) \leq L \cdot \mathsf{diam}\left[\frac{16\sqrt{d}}{n\sqrt{\rho}}\sqrt{\log(2/\beta)} + \frac{\sqrt{2}}{\sqrt{T}}\right].$$

*Proof.* The privacy proof is based on the zCDP analysis of the Gaussian mechanism and the composition property of zCDP [BS16].

Let $g_t \triangleq \nabla\hat{L}_n(\theta_t) + \xi_t$ and $\theta^\star \in \arg\min_{\theta\in\Theta}\hat{L}_n(\theta)$. Note that we can replace $\nabla\hat{L}_n(\theta_t)$ by any subgradient at $\theta_t$. By the convexity of $\ell$ and the first-order convexity condition we can write

$$\hat{L}_n\left(\frac{1}{T}\sum_{t\in[T]}\theta_t\right) - \hat{L}_n(\theta^\star) \leq \frac{1}{T}\sum_{i\in[T]}\hat{L}_n(\theta_t) - \hat{L}_n(\theta^\star)$$

$$\leq \frac{1}{T}\sum_{t\in[T]}\left\langle\nabla\hat{L}_n(\theta_t), \theta_t - \theta^\star\right\rangle.$$

Then, by the contraction property of the projection, we can write

$$\|\theta_{t+1} - \theta^\star\|^2 = \|\Pi_\Theta(\theta_t - \eta g_t) - \theta^\star\|^2$$

$$\leq \|\theta_t - \theta^\star - \eta g_t\|^2$$

$$= \|\theta_t - \theta^\star\|^2 + \eta^2\|g_t\|^2 - 2\eta\langle g_t, \theta_t - \theta^\star\rangle$$

$$\leq \|\theta_t - \theta^\star\|^2 + 2\eta^2\left(\left\|\nabla\hat{L}_n(\theta_t)\right\|^2 + \|\xi_t\|^2\right) - 2\eta\langle g_t, \theta_t - \theta^\star\rangle$$

$$\leq \|\theta_t - \theta^\star\|^2 + 2\eta^2 L^2 + 2\eta^2\|\xi_t\|^2 - 2\eta\langle g_t, \theta_t - \theta^\star\rangle.$$

Here, we have used for every $a, b \in \mathbb{R}^d$, $\|a + b\|^2 \leq 2\|a\|^2 + 2\|b\|^2$, and $\left\|\nabla\hat{L}_n(\theta)\right\| \leq L$ for every $\theta$. Therefore, we conclude that

$$\langle g_t, \theta_t - \theta^\star\rangle \leq \frac{1}{2\eta}\left(\|\theta_t - \theta^\star\|^2 - \|\theta_{t+1} - \theta^\star\|^2\right) + \eta\|\xi_t\|^2 + \eta L^2. \tag{4}$$

Define the following random variable for every $t \in [T]$

$$Y_t = \left\langle \nabla \hat{L}_n(\theta_t), \theta_t - \theta^\star \right\rangle - \langle g_t, \theta_t - \theta^\star \rangle. \tag{5}$$

Also, define the following filtration

$$\mathcal{F}_t = \sigma(\theta_0, \dots, \theta_t), \tag{6}$$

which is the sigma-field generated by $\theta_0, \dots, \theta_t$.

**Lemma A.2.** $\{Y_t\}_{t \in [T]}$ *is a martingale difference sequence adapted to* $\{\mathcal{F}_t\}_{t \in [T]}$.

*Proof.* Notice that $\nabla \hat{L}_n(\theta_t)$, $\theta_t$, and $\theta^\star$ are $\mathcal{F}_t$-measurable. Therefore, we can write

$$\mathbb{E}[Y_t | \mathcal{F}_t] = \mathbb{E}[\langle g_t, \theta_t - \theta^\star \rangle - \left\langle \nabla \hat{L}_n(\theta_t), \theta_t - \theta^\star \right\rangle | \mathcal{F}_t]$$

$$= \langle \mathbb{E}[g_t | \mathcal{F}_t], \theta_t - \theta^\star \rangle - \left\langle \nabla \hat{L}_n(\theta_t), \theta_t - \theta^\star \right\rangle.$$

By definition $\xi_t$ is independent of the history up to time $t$. Therefore, $\mathbb{E}[\xi_t | \mathcal{F}_t] = 0$ since $\mathbb{E}[\xi_t] = 0$ which gives

$$\mathbb{E}[g_t | \mathcal{F}_t] = \mathbb{E}[\nabla \hat{L}_n(\theta_t) + \xi_t | \mathcal{F}_t] = \nabla \hat{L}_n(\theta_t), \tag{7}$$

Therefore, $\mathbb{E}[Y_t | \mathcal{F}_t] = 0$. Moreover, by Cuachy-Schwartz inequality and the boundedness of $\Theta$ we can write

$$\mathbb{E}[|Y_t|] = \mathbb{E}[|\langle \xi_t, \theta_t - \theta^\star \rangle|] \leq \mathbb{E}[\|\xi_t\| \|\theta_t - \theta^\star\|] \leq R \mathbb{E}[\|\xi_t\|] < \infty. \tag{8}$$

Therefore, $\{Y_t\}_{t \in [T]}$ is a martingale difference sequence as was to be shown. $\square$

Using Equation (4) and by the definition of $Y_t$ in Equation (5), we can write

$$\left\langle \nabla \hat{L}_n(\theta_t), \theta_t - \theta^\star \right\rangle \leq \frac{1}{2\eta} \left( \|\theta_t - \theta^\star\|^2 - \|\theta_{t+1} - \theta^\star\|^2 \right) + \eta \|\xi_t\|^2 + \eta L^2 + Y_t.$$

Summing it from 0 to $T - 1$ gives

$$\frac{1}{T} \sum_{t \in [T]} \left\langle \nabla \hat{L}_n(\theta_t), \theta_t - \theta^\star \right\rangle \leq \frac{1}{2\eta T} \|\theta_0 - \theta^\star\|^2 + \frac{\eta}{T} \sum_{t \in [T]} \|\xi_t\|^2 + \eta L^2 + \frac{1}{T} \sum_{t \in [T]} Y_t$$

$$\leq \frac{R^2}{2\eta T} + \eta L^2 + \underbrace{\frac{\eta}{T} \sum_{t \in [T]} \|\xi_t\|^2}_{(A)} + \underbrace{\frac{1}{T} \sum_{t \in [T]} Y_t}_{(B)}. \tag{9}$$

**Analyzing (A) in Equation (9)**

Notice that $\sum_{t \in [T]} \|\xi_t\|^2 \overset{d}{=} \sigma^2 \|Y\|^2$. Therefore, for every $\beta \in (0, 1)$ provided that $Td \geq \log(4/\beta)$, with probability at least $1 - \beta/2$, we have

$$\frac{\eta}{T} \sum_{t \in [T]} \|\xi_t\|^2 \leq \eta \sigma^2 d \left( 1 + 4 \sqrt{\frac{\log(2/\beta)}{Td}} \right). \tag{10}$$

**Analyzing (B) in Equation (9)**

**Lemma A.3** (Shamir [Sha11])**.** *Let* $m \in \mathbb{N}$. *Let* $\{Z_m\}_{m \in [M]}$ *be a martingale difference sequence adapted to a filtration* $\{\mathcal{F}_m\}_{m \in [M]}$, *and suppose there are constants* $b > 1$ *and* $c > 0$ *such that for any* $m$ *and any* $\alpha > 0$, *it holds that*

$$\mathbb{P}(|Z_t| \geq \alpha | \mathcal{F}_t) \leq b \exp(-c\alpha^2).$$

*Then for any* $\beta > 0$, *it holds with probability at least* $1 - \beta$ *that*

$$\frac{1}{M} \sum_{m \in [M]} Z_m \leq \sqrt{\frac{28b \log(1/\beta)}{cM}}.$$

We can rephrase *Equation* (5) as

$$Y_t = \left\langle \nabla \hat{L}_n(\theta_t), \theta_t - \theta^\star \right\rangle - \langle g_t, \theta_t - \theta^\star \rangle = \langle \xi_t, \theta^\star - \theta_t \rangle.$$

Notice that condition on $\mathcal{F}_t$, $\langle \xi_t, \theta^\star - \theta_t \rangle | \mathcal{F}_t \sim \mathcal{N}(0, \sigma^2 \|\theta_t - \theta^\star\|^2)$. Therefore,

$$\mathbb{P}(|\langle \xi_t, \theta^\star - \theta_t \rangle| \geq \alpha | \mathcal{F}_t) \leq 2 \exp\left(-\frac{\alpha^2}{2\sigma^2 \|\theta_t - \theta^\star\|^2}\right) \leq 2 \exp\left(-\frac{\alpha^2}{2\sigma^2 R^2}\right).$$

Note that the bound holds for every $t \in [T]$. Therefore, using Lemma A.3, with probability at least $1 - \beta/2$, we have

$$\frac{1}{T} \sum_{t \in [T]} \langle \xi_t, \theta^\star - \theta_t \rangle \leq 2\sigma R \sqrt{\frac{28 \log(2/\beta)}{T}}. \tag{11}$$

From Equation (9), Equation (10), and Equation (11), we have with probability at least $1 - \beta$

$$\hat{L}_n\left(\frac{1}{T} \sum_{t \in [T]} \theta_t\right) - \min_{\theta \in \Theta} \hat{L}_n(\theta) \leq \frac{R^2}{2\eta T} + \eta L^2 + \eta \sigma^2 d\left(1 + 4\sqrt{\frac{\log(4/\beta)}{Td}}\right) + 2\sigma R \sqrt{\frac{28 \log(2/\beta)}{T}}, \tag{12}$$

provided that $Td \geq \log(4/\beta)$. Let

$$\sigma^2 = \frac{L^2 T}{2\rho n^2} \quad, \quad \eta = \frac{R}{L\sqrt{T}} \cdot \frac{1}{\sqrt{2 + \frac{5dT}{\rho n^2}}}.$$

Using these parameters, we obtain that

$$\hat{L}_n\left(\frac{1}{T} \sum_{t \in [T]} \theta_t\right) - \min_{\theta \in \Theta} \hat{L}_n(\theta) \leq \frac{RL\sqrt{d}}{n\sqrt{\rho}} \left[\sqrt{1 + \frac{2\rho n^2}{Td}} + \sqrt{56 \log(2/\beta)}\right]$$

$$\leq \frac{2RL\sqrt{d}}{n\sqrt{\rho}} \sqrt{56 \log(2/\beta)} + \frac{RL\sqrt{2}}{\sqrt{T}}, \tag{13}$$

where the last step is by assuming that $1 \leq \sqrt{56 \log(2/\beta)}$.

$\square$

## B  Above Threshold Algorithm

---
**Algorithm 7** `AboveThreshold`

---
1: Inputs: Queries $\{f_0, \ldots, f_{k-1}\}$, Privacy Budget $\rho$-zCDP, Threshold $T$.
2: $\xi_{\text{tresh}} \sim \text{Lap}\left(\frac{6}{\sqrt{2\rho}}\right)$
3: $\hat{T} = T + \xi_{\text{tresh}}$
4: **for** $i \in [k]$ **do**
5:    $\xi_i \sim \text{Lap}\left(\frac{12}{\sqrt{2\rho}}\right)$
6:    **if** $f_i + \xi_i > \hat{T}$: **then**
7:       Output $\hat{\Delta} = i$.
8:       Halt
9: Output `Fail`.

---

## C  Technical Lemma

**Lemma C.1.** *Let $\sigma > 0$. Let $Y$ be a random variable with the distribution $\mathcal{N}(0, \sigma^2)$. Then, for every $\beta \in (0, 1]$, we have $\mathbb{P}\left(|Y| > \sigma\sqrt{2\log(2/\beta)}\right) \leq \beta$.*

**Lemma C.2** (Laurent and Massart [LM00])**.** *Let $m \in \mathbb{N}$. Consider random vector $Y \sim \mathcal{N}(0, \mathbb{I}_m)$. Then, for every $t \geq 0$,*

$$\mathbb{P}\left( \|Y\|^2 \geq m + 2\sqrt{tm} + 2t \right) \leq \exp(-t)$$

**Corollary C.3.** *Let $\beta \in (0, 1)$, $m \in \mathbb{N}$, and $m \geq \log \frac{2}{\beta}$. Consider $Y \sim \mathcal{N}(0, \mathbb{I}_m)$, then*

$$\mathbb{P}\left( m\left( 1 - 2\sqrt{\frac{\log(2/\beta)}{m}} \right) \leq \|Y\|^2 \leq m\left( 1 + 4\sqrt{\frac{\log(2/\beta)}{m}} \right) \right) \geq 1 - \beta,$$

**Lemma C.4.** *Let $n \in \mathbb{N}$ and $n \geq 4$. Let $\mathbf{X}^{(n)} \in (\mathbb{R}^d)^n$ and $\tilde{\mathbf{X}}^{(n)} \in (\mathbb{R}^d)^n$ be two datasets that differ in one sample. Let $\theta^\star \in \mathrm{GM}(\mathbf{X}^{(n)})$ and $\theta^\circledast \in \mathrm{GM}(\tilde{\mathbf{X}}^{(n)})$. Let $\Delta_{3n/4}(\theta^\star)$ be the radius of the ball around $\theta^\star$ that contains at least $3n/4$ of $\mathbf{X}^{(n)}$. Then,*

$$\left\| \theta^\circledast - \theta^\star \right\| \leq \frac{3}{2}\Delta_{3n/4}(\theta^\star).$$

*Proof.* The proof is by contrapositive. In particular, we show that for every $\theta \in \mathbb{R}^d$ such that $\|\theta - \theta^\star\| > \frac{3}{2}\Delta_{3n/4}(\theta^\star)$, we have, $\theta \notin \mathrm{GM}(\tilde{\mathbf{X}}^{(n)})$. Let $\mathcal{I} = \{i \in [n] : x_i \in \mathcal{B}_d(\theta^\star, \Delta_{3n/4}(\theta^\star))$ and $x_i \in \tilde{\mathbf{X}}^{(n)}\}$. Using the variational representation of $\|\cdot\|_2$, we can write

$$
\begin{aligned}
\left\| \nabla F(\theta; \tilde{\mathbf{X}}^{(n)}) \right\| &\geq \left\langle \nabla F(\theta; \tilde{\mathbf{X}}^{(n)}), \frac{\theta - \theta^\star}{\|\theta - \theta^\star\|} \right\rangle \\
&= \sum_{i \in \mathcal{I}} \left\langle \frac{\theta - x_i}{\|\theta - x_i\|}, \frac{\theta - \theta^\star}{\|\theta - \theta^\star\|} \right\rangle + \sum_{i \in [n] \setminus \mathcal{I}} \left\langle \frac{\theta - x_i}{\|\theta - x_i\|}, \frac{\theta - \theta^\star}{\|\theta - \theta^\star\|} \right\rangle \\
&\geq \sum_{i \in \mathcal{I}} \left\langle \frac{\theta - x_i}{\|\theta - x_i\|}, \frac{\theta - \theta^\star}{\|\theta - \theta^\star\|} \right\rangle - (n - |\mathcal{I}|)
\end{aligned}
$$

where the last step follows from Cauchy–Schwarz inequality. Then, we can write

$$
\begin{aligned}
\left\| \nabla F(\theta; \tilde{\mathbf{X}}^{(n)}) \right\| &\geq |\mathcal{I}|\sqrt{1 - \left( \frac{\Delta_{3n/4}(\theta^\star)}{\|\theta - \theta^\star\|} \right)^2} - (n - |\mathcal{I}|) \\
&= |\mathcal{I}|\left( 1 + \sqrt{1 - \left( \frac{\Delta_{3n/4}(\theta^\star)}{\|\theta - \theta^\star\|} \right)^2} \right) - n \\
&\geq (3n/4)\left( 1 + \sqrt{1 - \left( \frac{\Delta_{3n/4}(\theta^\star)}{\|\theta - \theta^\star\|} \right)^2} \right) - n \\
&\geq (3n/4)\left( 1 + \sqrt{1 - 4/9} \right) - n \\
&> 0.
\end{aligned}
\tag{14}
$$

Therefore $\|\theta^\circledast - \theta^\star\| \leq 3/2\Delta_{3n/4}(\theta^\star)$. $\qquad\square$

**Lemma C.5.** *For every $n \in \mathbb{N}$ and for every $\mathbf{X}^{(n)} = (x_1, \ldots, x_n)$, we have $GM(\mathbf{X}^{(n)})$ lies in the convex hull of $\{x_1, \ldots, x_n\}$.*

**Lemma C.6.** *Let $(x_1, \ldots, x_n) \in (\mathbb{R}^d)^n$ be a dataset and $\theta^\star = \mathrm{GM}((x_1, \ldots, x_n))$. Let $\mathcal{B} \subseteq [n]$ such that $|\mathcal{B}| < n/2$. Then, for every $\theta$, we have*

$$\|\theta - \theta^\star\| \leq \frac{2n - 2|\mathcal{B}|}{n - 2|\mathcal{B}|} \max_{i \notin \mathcal{B}} \|\theta - x_i\|.$$

*Proof.* It is a simple modification of [CLMPS16, Lemma. 24]. $\qquad\square$

# D  Proof of Section 2

*Proof of Lemma 2.2.* The proof follows closely [NSV16, Lemma 4.5]. Let $\mathbf{X}$ and $\mathbf{X}'$ are two neighboring datasets of size $n$ that differ in the first sample. For a fixed $\nu$ and $i \in [n]$, if $i \neq 1$, $N_i(\nu)$ can change by one. Also, in the worst-case the new datapoint can be close to the rest of the datapoints. Therefore, the sensitivity is bounded by $\frac{1}{m}((m-1)+n) \leq 1 + \frac{n}{m} \leq 1 + \frac{1}{\gamma} \leq 3$ where the last step follows from $\gamma \geq 1/2$. $\qquad\square$

*Proof of Lemma 2.3.* Let $\hat{\nu}$ be such that $N(\hat{\nu}) \geq \lceil \gamma_1 n \rceil$, by definition of $N(\cdot)$ it means that there exists a datapoint $x_i$ such that the ball of radius $\hat{\nu}$ around it contains at least $\lceil \gamma_1 n \rceil$ datapoints. Let $\mathcal{B} = \{j \in [n] : \|x_i - x_j\| > \hat{\nu}\}$. By the described argument, we have $|\mathcal{B}| \leq (1 - \gamma_1)n$. Then, we invoke Lemma C.6 with the described $\mathcal{B}$ and $\theta = x_i$ to obtain

$$\|\theta^\star - x_i\| \leq \frac{2n - 2(1-\gamma_1)n}{n - 2(1-\gamma_1)n}\hat{\nu}$$
$$= \frac{2\gamma_1}{2\gamma_1 - 1}\hat{\nu}.$$

The first step follows because function $h : \mathbb{R} \to \mathbb{R}, h(z) = \frac{2n - 2z}{n - 2z}$ is increasing for $z < n/2$. In the next step, we use the triangle inequality to write

$$\Delta_{\gamma_1 n}(\theta^\star) \leq \Delta_{\gamma_1 n}(x_i) + \|x_i - \theta^\star\|$$
$$\leq \hat{\nu} + \frac{2\gamma_1}{2\gamma_1 - 1}\hat{\nu}$$
$$= \frac{4\gamma_1 - 1}{2\gamma_1 - 1}\hat{\nu},$$

where $\Delta_{\cdot}(\cdot)$ is defined in Definition 1.1.

Next, we turn into proving the upperbound on $\hat{\nu}$. By assumption $N(\hat{\nu}/2) < \lceil \gamma_2 n \rceil$. For the sake of contradiction, assume that $\hat{\nu} > 4\Delta_{\gamma_2 n}(\theta^\star)$. Then, consider the set $\mathcal{G} = \{i \in [n] : \|\theta^\star - x_i\| \leq \Delta_{\gamma_2 n}(\theta^\star)\}$. By definition, $|\mathcal{G}| \geq \lceil \gamma_2 n \rceil$. Consider an arbitrary subset of $\mathcal{G}$ with the size of $\lceil \gamma_2 n \rceil$. The main observation, which follows from the triangle inequality, is that a ball of radius $2\Delta_{\gamma_2 n}(\theta^\star)$ around every point in $\mathcal{G}$ contains at least $\lceil \gamma_2 n \rceil$ datapoint. Therefore, $N(\hat{\nu}/2) \geq N(2\Delta_{\gamma_2 n}(\theta^\star)) \geq \lceil \gamma_2 n \rceil$ which contradicts with the assumption that $N(\hat{\nu}/2) < \lceil \gamma_2 n \rceil$. $\qquad\square$

*Proof of Theorem 2.4.* The privacy proof simply follows from the privacy analysis in [DR+14, Sec. 3.6]. We focus here on the utility guarantees.

**Part 1:**  Let $k = \lceil \log\left(\frac{2R}{r}\right) \rceil$. It is simple to see that

$$\mathbb{P}\left(\hat{\Delta} = \texttt{Fail}\right) \leq \mathbb{P}\left(N(2^k r) + \xi_k \leq m + \xi_{\text{thresh}}\right)$$
$$= \mathbb{P}(n - m \leq \xi_{\text{thresh}} - \xi_k)$$
$$\leq \mathbb{P}((1-\gamma)n \leq \xi_{\text{thresh}} - \xi_k)$$

where the last step follows from the assumption that $\max_{x_i, x_j \in \mathbf{X}^{(n)}} \|x_i - x_j\| \leq 2R$, which gives us $N(2^k r) = n$ by the definition of $N(\cdot)$. By a simple tail bound on the Laplace distribution, we have $\mathbb{P}\left(|\xi_{\text{thresh}}| \geq \frac{6}{\sqrt{2\rho}} \log(4/\beta)\right) \leq \beta/4$ and $\mathbb{P}\left(|\xi_k| \geq \frac{12}{\sqrt{2\rho}} \log(4/\beta)\right) \leq \beta/4$. Therefore, given $n > \frac{1}{1-\gamma} \frac{18}{\sqrt{2\rho}} \log(4/\beta)$, $\mathbb{P}(n - m \leq \xi_{\text{thresh}} - \xi_k) \leq \beta/2$.

**Part 2:**  Lemma C.6 implies that for every $\nu$ such that $N(\nu) \geq \lceil \gamma n \rceil$, we have, $\Delta_{\gamma n}(\theta^\star) \cdot \frac{2\gamma - 1}{4\gamma - 1} \leq \nu$. Therefore, we can write

$$\mathbb{P}\left(\Delta_{\gamma n}(\theta^\star)\frac{2\gamma-1}{4\gamma-1} \leq \hat{\Delta}\right) \geq \mathbb{P}\left(N\left(\hat{\Delta}\right) \geq \lceil \gamma n \rceil\right) \Leftrightarrow \mathbb{P}\left(\Delta_{\gamma n}(\theta^\star)\frac{2\gamma-1}{4\gamma-1} > \hat{\Delta}\right) \leq \mathbb{P}\left(N\left(\hat{\Delta}\right) < \lceil \gamma n \rceil\right).$$

Consider

$$\mathbb{P}\left(N\left(\hat{\Delta}\right) < \lceil \gamma n \rceil\right) \leq \mathbb{P}\left(N\left(\hat{\Delta}\right) < \lceil \gamma n \rceil \text{ and } \hat{\Delta} \neq \texttt{Fail}\right) + \mathbb{P}\left(\hat{\Delta} = \texttt{Fail}\right). \qquad (15)$$

Under the event that $\hat{\Delta} \neq \texttt{Fail}$, there exists $i \in \{0, \ldots, k-1\}$, such that

$$N(\hat{\Delta}) + \xi_i \geq m + \frac{18}{\sqrt{2\rho}} \log(2(k+1)/\beta) + \xi_{\text{thresh}}$$

By a simple tail bound and union bound, we have

$$
\begin{aligned}
\mathbb{P}(\mathcal{B}) &\triangleq \mathbb{P}\left( |\xi_{\text{thresh}}| \geq \frac{6}{\sqrt{2\rho}} \log(2(k+1)/\beta) \text{ and } \{\max_i |\xi_i| \geq \frac{12}{\sqrt{2\rho}} \log(2(k+1)/\beta)\} \right) \\
&\leq \beta/2,
\end{aligned}
\tag{16}
$$

where $k = \lceil \log\left(\frac{2R}{r}\right) \rceil$. We further upperbound the first term in Equation (15) as follows

$$\mathbb{P}\left( N\left(\hat{\Delta}\right) < \lceil \gamma n \rceil \text{ and } \hat{\Delta} \neq \texttt{Fail} \right) \leq \mathbb{P}\left( N\left(\hat{\Delta}\right) < \lceil \gamma n \rceil \text{ and } \hat{\Delta} \neq \texttt{Fail} \text{ and } \mathcal{B}^c \right) + \mathbb{P}(\mathcal{B}).$$

We claim that the first term in this equation is zero. Recall that $m = \lceil \gamma n \rceil$. Under the event $\mathcal{B}^c$, $\xi_{\text{thresh}} - \xi_i \geq -\frac{18}{\sqrt{2\rho}} \log(2(k+1)/\beta)$ and as a result, $m + \frac{18}{\sqrt{2\rho}} \log(2(k+1)/\beta) + \xi_{\text{thresh}} - \xi_i \geq m$. Therefore, it shows that the probability of the first term is zero. Also, as showed above, $\mathbb{P}(\mathcal{B}) \leq \beta/2$. Therefore, $\mathbb{P}\left( N\left(\hat{\Delta}\right) < \lceil \gamma n \rceil \right) \leq \mathbb{P}(\mathcal{B}) + \mathbb{P}\left( \hat{\Delta} = \texttt{Fail} \right)$. Combining it with $\mathbb{P}\left( \hat{\Delta} = \texttt{Fail} \right) \leq \beta/2$ concludes the proof.

**Part 3:** Assume that $N(r) < m$. Let $k = \lceil \log\left(\frac{2R}{r}\right) \rceil$. Let $\tilde{\gamma} = \gamma + \frac{1}{n} \frac{18}{\sqrt{2\rho}} \log(2(k+1)/\beta)$. In Part 2, we showed that given $n > \frac{18}{\sqrt{2\rho}} \log(4/\beta)$, we have

$$\mathbb{P}\left( \Delta_{\gamma n}(\theta^\star) \frac{2\gamma - 1}{4\gamma - 1} \leq \hat{\Delta} \right) \geq 1 - \beta.$$

We only focus on the upperbound. From Lemma 2.3, we have

$$\mathbb{P}\left( \hat{\Delta} \leq 4\Delta_{\tilde{\gamma} n}(\theta^\star) \right) \geq \mathbb{P}\left( N\left(\hat{\Delta}/2\right) \leq \lceil \tilde{\gamma} n \rceil \right) \Leftrightarrow \mathbb{P}\left( \hat{\Delta} > 4\Delta_{\tilde{\gamma} n}(\theta^\star) \right) \leq \mathbb{P}\left( N\left(\hat{\Delta}/2\right) > \lceil \tilde{\gamma} n \rceil \right). \tag{17}$$

We can write

$$
\begin{aligned}
\mathbb{P}\left( N\left(\hat{\Delta}/2\right) > \lceil \tilde{\gamma} n \rceil \right) &\leq \mathbb{P}\left( N\left(\hat{\Delta}/2\right) > \lceil \tilde{\gamma} n \rceil \text{ and } \hat{\Delta} \notin \{r, \texttt{Fail}\} \right) + \mathbb{P}\left( \hat{\Delta} \in \{r, \texttt{Fail}\} \right) \\
&\leq \mathbb{P}\left( N\left(\hat{\Delta}/2\right) > \lceil \tilde{\gamma} n \rceil \text{ and } \hat{\Delta} \notin \{r, \texttt{Fail}\} \right) + \mathbb{P}\left( \hat{\Delta} = r \right) + \mathbb{P}\left( \hat{\Delta} = \texttt{Fail} \right),
\end{aligned}
$$

where the last step follows from the union bound. For the first term, we have

$$
\begin{aligned}
&\mathbb{P}\left( N\left(\hat{\Delta}/2\right) > \lceil \tilde{\gamma} n \rceil \text{ and } \hat{\Delta} \notin \{r, \texttt{Fail}\} \right) \\
&= \mathbb{P}\left( N\left(\hat{\Delta}/2\right) > \lceil \tilde{\gamma} n \rceil \text{ and } \hat{\Delta} \notin \{r, \texttt{Fail}\} \text{ and } N(\hat{\Delta}/2) + \xi_{\hat{i}} < m + \frac{18}{\sqrt{\rho}} \log\left( \frac{2}{\beta} \cdot \left\lceil \log\left(\frac{2R}{r}\right) \right\rceil \right) + \xi_{\text{thresh}} \right),
\end{aligned}
\tag{18}
$$

where $\hat{i} = \log\left( \hat{\Delta}/2r \right) - 1$. The last step follows from the following observation: under the event that $\hat{\Delta} \notin \{r, \texttt{Fail}\}$, during the execution of Algorithm 7, both $N(\hat{\Delta})$ and $N(\hat{\Delta}/2)$ are compared to the noisy threshold. Using the tail bounds in Equation (16), we have under the event $\mathcal{B}^c$, with probability at least $1 - \beta/2$,

$$
\begin{aligned}
m + \frac{18}{\sqrt{2\rho}} \log\left( \frac{2}{\beta} \cdot \left\lceil \log\left(\frac{2R}{r}\right) \right\rceil \right) + \xi_{\text{thresh}} - \xi_{\hat{i}} &\leq m + \frac{18}{\sqrt{2\rho}} \log\left( \frac{2}{\beta} \cdot \left\lceil \log\left(\frac{2R}{r}\right) \right\rceil \right) + \frac{18}{\sqrt{2\rho}} \log(2(k+1)/\beta) \\
&\leq m + \frac{36}{\sqrt{2\rho}} \log(2(k+1)/\beta).
\end{aligned}
$$

This shows that we have

$$\mathbb{P}\left( N\left(\hat{\Delta}/2\right) > \lceil \tilde{\gamma} n \rceil \text{ and } \hat{\Delta} \notin \{r, \texttt{Fail}\} \text{ and } N(\hat{\Delta}/2) + \xi_{\hat{i}} < m + \frac{18}{\sqrt{\rho}} \log\left( \frac{2}{\beta} \cdot \left\lceil \log\left(\frac{2R}{r}\right) \right\rceil \right) + \xi_{\text{thresh}} \right) \leq \beta/2.$$

In the next step, we bound $\mathbb{P}\left(\hat{\Delta} = r\right)$. Notice that

$$\mathbb{P}\left(\hat{\Delta} = r\right) = \mathbb{P}\left(N(r) + \xi_1 > m + \frac{18}{\sqrt{2\rho}}\log\left(\frac{2}{\beta} \cdot \left\lceil\log\left(\frac{2R}{r}\right)\right\rceil\right) + \xi_{\text{thresh}}\right).$$

Using simple tail bound, we have $\mathbb{P}\left(\xi_{\text{thresh}} - \xi_1 \leq -\frac{18}{\sqrt{2\rho}}\log\left(\frac{2}{\beta} \cdot \left\lceil\log\left(\frac{2R}{r}\right)\right\rceil\right)\right) \leq \beta/2$ which shows that $\mathbb{P}\left(\hat{\Delta} = r\right) \leq \beta/2$ since we assume that $N(r) < m$. Therefore, combining all the pieces together, we proved

$$\mathbb{P}\left(\Delta_{\gamma n}(\theta^\star)\frac{2\gamma - 1}{4\gamma - 1} \leq \hat{\Delta} \leq 4\Delta_{\tilde{\gamma} n}(\theta^\star)\right) \geq 1 - \frac{5\beta}{2}.$$

**Part 4:** In Part 2, we showed that given $n > \frac{18}{(1-\gamma)\sqrt{2\rho}}\log(4/\beta)$, we have

$$\mathbb{P}\left(\hat{\Delta} \leq \Delta_{\gamma n}(\theta^\star)\frac{2\gamma - 1}{4\gamma - 1}\right) \leq \beta. \tag{19}$$

Consider the following event $\mathcal{E} = \left\{\hat{\Delta} \leq 4\Delta_{\tilde{\gamma} n}(\theta^\star) \text{ or } \hat{\Delta} = r\right\}$. We have

$$\begin{aligned}
\mathbb{P}(\mathcal{E}^c) &= \mathbb{P}\left(\hat{\Delta} > 4\Delta_{\tilde{\gamma} n}(\theta^\star) \text{ and } \hat{\Delta} \neq r\right) \\
&\leq \mathbb{P}\left(\hat{\Delta} > 4\Delta_{\tilde{\gamma} n}(\theta^\star) \text{ and } \hat{\Delta} \neq \{r, \mathsf{Fail}\}\right) + \mathbb{P}\left(\hat{\Delta} = \mathsf{Fail}\right) \\
&\leq \mathbb{P}\left(N\left(\hat{\Delta}/2\right) > \lceil\tilde{\gamma} n\rceil \text{ and } \hat{\Delta} \neq \{r, \mathsf{Fail}\}\right) + \mathbb{P}\left(\hat{\Delta} = \mathsf{Fail}\right).
\end{aligned}$$

Here, the last step follows from Equation (17). Notice that in Equation (18), we analyzed the probability of the first term and we showed that it is as most $\beta/2$. We also have that $\mathbb{P}\left(\hat{\Delta} = \mathsf{Fail}\right) \leq \beta/2$ from Part 1. Therefore, $\mathbb{P}(\mathcal{E}^c) \leq \beta$. Combining it with Equation (19) concludes the proof. $\qquad\square$

*Proof of Lemma 2.6.* Let $\mathcal{I} = \{i \in [n] : \|\theta_0 - x_i\| \leq \Delta_{\gamma n}(\theta_0)\}$. For every $i \in \mathcal{I}$, we have $\|x_i - \theta_0\| \leq \Delta_{\gamma n}(\theta_0)$. Using the triangle inequality, for every $i \in \mathcal{I}$, we can write

$$\begin{aligned}
\|\theta_1 - x_i\| &\geq \|\theta_1 - \theta_0\| - \|\theta_0 - x_i\| \\
&\geq \|\theta_1 - \theta_0\| - (2\Delta_{\gamma n}(\theta_0) - \|\theta_0 - x_i\|).
\end{aligned}$$

The last equation is equivalent to

$$\|\theta_1 - x_i\| - \|\theta_0 - x_i\| \geq \|\theta_1 - \theta_0\| - 2\Delta_{\gamma n}(\theta_0). \tag{20}$$

Then, for every $i \notin \mathcal{I}$, by an application of the triangle inequality

$$\begin{aligned}
\|\theta_1 - x_i\| + \|\theta_1 - \theta_0\| &\geq \|\theta_0 - x_i\| \\
(\Leftrightarrow)\|\theta_1 - x_i\| - \|\theta_0 - x_i\| &\geq -\|\theta_1 - \theta_0\|.
\end{aligned} \tag{21}$$

Then, by adding both sides of Equation (20) and Equation (21), we have

$$F(\theta_1; \mathbf{X}^{(n)}) - F(\theta_0; \mathbf{X}^{(n)}) \geq |\mathcal{I}|\|\theta_1 - \theta_0\| - (n - |\mathcal{I}|)\|\theta_1 - \theta_0\| - 2|\mathcal{I}|\Delta_{\gamma n}(\theta_0).$$

This equation can be represented as

$$\begin{aligned}
\|\theta_1 - \theta_0\| &\leq \frac{F(\theta_1; \mathbf{X}^{(n)}) - F(\theta_0; \mathbf{X}^{(n)}) + 2|\mathcal{I}|\Delta_{\gamma n}(\theta_0)}{2|\mathcal{I}| - n} \\
&\leq \frac{\zeta n + 2|\mathcal{I}|\Delta_{\gamma n}(\theta_0)}{2|\mathcal{I}| - n}.
\end{aligned} \tag{22}$$

Let $\gamma' n = |\mathcal{I}|$. We know that $\gamma' \geq \gamma$. Using this representation we can write

$$\|\theta_1 - \theta_0\| \leq \frac{\zeta + 2\gamma'\Delta_{\gamma n}(\theta_0)}{2\gamma' - 1}.$$

For a fixed $a, b > 0$ define $h(x) \triangleq \frac{a + 2xb}{2x - 1}$. For $x > 1/2$, $\frac{\mathrm{d}h(x)}{\mathrm{d}x} = -\frac{2(a+b)}{(2x-1)^2}$. This shows that $h(x)$ is decreasing for $x > 1/2$. Therefore using this observation

$$\frac{\zeta + 2\gamma'\Delta_{\gamma n}(\theta_0)}{2\gamma' - 1} \leq \frac{\zeta + 2\gamma\Delta_{\gamma n}(\theta_0)}{2\gamma - 1},$$

as was to be shown. $\qquad\square$

*Proof of Theorem 2.7.* The privacy proof is straightforward. Algorithm 2 uses the data in Line 3 and Line 11. Based on the privacy budget allocation and the composition properties of zCDP, we can show that the output satisfies $\rho$-zCDP.

For the claim regarding utility, in the first step, consider the recursion in Line 12 of Algorithm 2, i.e., $\text{rad}_{t+1} = \frac{1}{2}\text{rad}_t + 12\hat{\Delta}$ initialized at $\text{rad}_0 = R$. It can be easily shown that $\text{rad}_m = \frac{1}{2^m}\text{rad}_0 + 12\hat{\Delta}\sum_{i=0}^{m-1}(1/2)^i$ for $m \geq 1$. In particular, let $k_{\text{wu}} = \frac{1}{\log(2)}\log\big(R/\hat{\Delta}\big)$, then, we obtain that $\text{rad}_{k_{\text{wu}}} \leq 25\hat{\Delta}$.

Let $\gamma = 3/4$ and

$$\tilde{\gamma} = \gamma + \frac{1}{n}\frac{36\sqrt{2}}{\sqrt{2\rho}}\log\bigg(2\bigg(\bigg\lceil\log\bigg(\frac{2R}{r}\bigg)\bigg\rceil + 1\bigg)\frac{2}{\beta}\bigg) \leq 0.75 + 0.05 = 0.8,$$

where the last step follows because $n \geq \Omega\Big(\frac{1}{\sqrt{\rho}}\log((\lceil\log(R/r)\rceil + 1)/\beta)\Big)$. Then, define the following event

$$\mathcal{G}_1 = \Big\{\Delta_{0.75n}(\theta^{\star}) \leq 4\hat{\Delta} \text{ and } \Big\{\hat{\Delta} \leq 4\Delta_{0.8n}(\theta^{\star}) \text{ or } \hat{\Delta} = r\Big\}\Big\}.$$

In the next step, we analyze the probability that $\theta^{\star} \in \Theta_{\text{loc}}$. We claim that

$$\mathbb{P}\big(\theta^{\star} \in \Theta_{\text{loc}}\big|\mathcal{G}_1\big) \geq (1 - \beta/(2k_{\text{wu}}))^{k_{\text{wu}}}.$$

We prove this by induction. In particular, we claim that for every $m \in \{0, \ldots, k_{\text{wu}}\}$ we have $\mathbb{P}\big(\theta^{\star} \in \Theta_m\big|\mathcal{G}_1\big) \geq (1 - \beta/(2k_{\text{wu}}))^m$. Note that we $\Theta_{\text{loc}} = \Theta_{k_{\text{wu}}}$.

For the base case, by the assumption that the datapoints are in $\mathcal{B}_d(R)$, we have $\mathbb{P}(\theta^{\star} \in \Theta_0|\mathcal{G}_1) = \mathbb{P}(\theta^{\star} \in \Theta_0) = 1$ since $\Theta_0$ is trivially independent of every random variable. Then, we show the claim for $m \in \{1, \ldots, k_{\text{wu}}\}$ assuming the claim holds for $m - 1$. We can write

$$\begin{aligned}
\mathbb{P}\big(\theta^{\star} \in \Theta_m\big|\mathcal{G}_1\big) &= \mathbb{P}\big(\|\theta^{\star} - \theta_m\| \leq \text{rad}_m\big|\mathcal{G}_1\big) \\
&\geq \mathbb{P}\big(\|\theta^{\star} - \theta_m\| \leq \text{rad}_m\big|\theta^{\star} \in \Theta_{m-1} \text{ and } \mathcal{G}_1\big)\mathbb{P}\big(\theta^{\star} \in \Theta_{m-1}\big|\mathcal{G}_1\big).
\end{aligned} \tag{23}$$

We claim that

$$\mathbb{P}\big(\|\theta^{\star} - \theta_m\| \leq \text{rad}_m\big|\theta^{\star} \in \Theta_{m-1} \text{ and } \mathcal{G}_1\big) \geq \mathbb{P}\bigg(\frac{2\big(F(\theta_m; \mathbf{X}^{(n)}) - F(\theta^{\star}; \mathbf{X}^{(n)})\big)}{n} \leq \frac{1}{2}\text{rad}_{m-1}\Big|\theta^{\star} \in \Theta_{m-1} \text{ and } \mathcal{G}_1\bigg).$$

To show this lets instantiate Lemma 2.6 with $\theta_0 = \theta^{\star}$ and $\gamma = 3/4$ to obtain that for every $\theta \in \mathbb{R}^d$,

$$\|\theta^{\star} - \theta\| \leq \frac{2\big(F(\theta; \mathbf{X}^{(n)}) - F(\theta^{\star}; \mathbf{X}^{(n)})\big)}{n} + 3\Delta_{\gamma n}(\theta^{\star}).$$

Notice that conditioned on $\mathcal{G}_1$, we have $3\Delta_{\gamma n}(\theta^{\star}) \leq 12\hat{\Delta}$. This shows that $\frac{2\big(F(\theta_m; \mathbf{X}^{(n)}) - F(\theta^{\star}; \mathbf{X}^{(n)})\big)}{n} \leq \frac{1}{2}\text{rad}_{m-1}$ implies that $\|\theta^{\star} - \theta_m\| \leq \text{rad}_m$ conditioned on $\mathcal{G}_1$ by the definition of $\text{rad}_m$ in Line 12. In the next step, we invoke Lemma A.1. Conditioned on $\theta^{\star} \in \Theta_{m-1}$ and $\mathcal{G}_1$, with probability at least $1 - \frac{\beta}{2k_{\text{wu}}}$, we have

$$F(\theta_m; \mathbf{X}^{(n)}) - F(\theta^{\star}; \mathbf{X}^{(n)}) \leq 2\text{rad}_{m-1} \cdot \left[\frac{16\sqrt{2dk_{\text{wu}}}}{n\sqrt{\rho}}\sqrt{\log(4k_{\text{wu}}/\beta)} + \frac{\sqrt{2}}{\sqrt{T_{\text{wu}}}}\right].$$

Notice $k_{\text{wu}} \leq \frac{1}{\log(2)}\log(R/r)$ a.s. By setting $T_{\text{wu}} = 128$ and the bound on the sample size, we have $F(\theta_m; \mathbf{X}^{(n)}) - F(\theta^{\star}; \mathbf{X}^{(n)}) \leq \frac{\text{rad}_{m-1}}{2}$. Also, notice that the randomness in DPGD is independent of history. Therefore,

$$\mathbb{P}\bigg(\frac{2\big(F(\theta_m; \mathbf{X}^{(n)}) - F(\theta^{\star}; \mathbf{X}^{(n)})\big)}{n} \leq \frac{1}{2}\text{rad}_{m-1}\Big|\theta^{\star} \in \Theta_{m-1} \text{ and } \mathcal{G}_1\bigg) \geq 1 - \frac{\beta}{2k_{\text{wu}}}, \tag{24}$$

Therefore, combining Equations (23) and (24), we obtain

$$\mathbb{P}\big(\theta^{\star} \in \Theta_m\big|\mathcal{G}_1\big) \geq \bigg(1 - \frac{\beta}{2k_{\text{wu}}}\bigg)^m,$$

as was to be shown. From Theorem 2.4, given $n \geq \Omega\left(\frac{1}{\sqrt{\rho}} \log((\lceil \log(R/r) \rceil + 1)/\beta)\right)$, we have

$$\mathbb{P}(\mathcal{G}_1) \geq 1 - \beta.$$

Therefore,

$$\mathbb{P}\left(\theta^\star \in \Theta_{\text{loc}} \text{ and } \mathcal{G}_1\right) = \mathbb{P}\left(\theta^\star \in \Theta_{\text{loc}} | \mathcal{G}_1\right)\mathbb{P}(\mathcal{G}_1) \geq \left(1 - \frac{\beta}{2k_{\text{wu}}}\right)^{k_{\text{wu}}} \cdot (1 - \beta) \geq (1 - 2\beta). \quad (25)$$

This proves the first claim.

Regarding the second claim, define the following event

$$\mathcal{G}_2 = \left\{\Delta_{0.75n}(\theta^\star)\frac{1}{4} \leq \hat{\Delta} \leq 4\Delta_{0.8n}(\theta^\star)\right\}.$$

Notice that in the proof of $\mathbb{P}(\theta^\star \in \Theta_{\text{loc}})$ we only used the fact that with a high probability $\Delta_{\gamma n}(\theta^\star)\frac{1}{4} \leq \hat{\Delta}$. Since $\mathcal{G}_2 \subseteq \mathcal{G}_1$, we can write

$$\mathbb{P}\left(\theta^\star \in \Theta_{\text{loc}} \text{ and } \mathcal{G}_2\right) = \mathbb{P}\left(\theta^\star \in \Theta_{\text{loc}} | \mathcal{G}_2\right)\mathbb{P}(\mathcal{G}_2) \geq (1 - \frac{\beta}{2k_{\text{wu}}})^{k_{\text{wu}}} \cdot (1 - 5\beta/4) \geq 1 - 2\beta,$$

where the last step follows from Part 3 of Theorem 2.4 which states that $\mathbb{P}(\mathcal{G}_2) \geq 1 - 5\beta/4$. $\quad\square$

# E  Proof of Section 3

*Proof of Theorem 3.1.* For the privacy proof, notice that Algorithm 3 uses the training set in Line 2 and Line 5. By the privacy budget allocation and the composition properties of zCDP in [BS16], it is immediate to see that the output satisfies $\rho$-zCDP.

Next, we prove the utility properties. Define the following event

$$\mathcal{G}_1 = \left\{\theta^\star \in \Theta_{\text{loc}} \text{ and } \Delta_{0.75n}(\theta^\star) \leq 4\hat{\Delta} \text{ and } \left\{\hat{\Delta} \leq 4\Delta_{0.8n}(\theta^\star) \text{ or } \hat{\Delta} = r\right\}\right\}.$$

Also, by the non-negativity of $\|\cdot\|_2$, we have

$$F(\theta^\star; \mathbf{X}^{(n)}) = \sum_{i=1}^{n} \|\theta^\star - x_i\| \geq 0.2n\Delta_{0.8n}(\theta^\star).$$

Using this inequality, for every $\theta$, we can write

$$F\left(\theta; \mathbf{X}^{(n)}\right) - F\left(\theta^\star; \mathbf{X}^{(n)}\right) \leq O\left(\frac{\sqrt{d}}{\sqrt{\rho}}\sqrt{\log(1/\beta)}\right)\Delta_{0.8n}(\theta^\star)$$

$$\Rightarrow F\left(\hat{\theta}; \mathbf{X}^{(n)}\right) - F\left(\theta; \mathbf{X}^{(n)}\right) \leq O\left(\frac{\sqrt{d}}{n\sqrt{\rho}}\sqrt{\log(1/\beta)}\right)F\left(\theta^\star; \mathbf{X}^{(n)}\right). \quad (26)$$

Under the event that $\theta^\star \in \Theta_0$, we can invoke Lemma A.1 to write

$$\mathbb{P}\left(F\left(\hat{\theta}; \mathbf{X}^{(n)}\right) - F\left(\theta^\star; \mathbf{X}^{(n)}\right) \leq O\left(\frac{\sqrt{d}}{\sqrt{\rho}}\sqrt{\log(1/\beta)}\right) \cdot \hat{\Delta} \,\Big|\, \mathcal{G}_1\right) \geq 1 - \beta/2,$$

where it follows because the internal randomness of DPGD is independent of the randomness in Localization step.

By the definition of event $\mathcal{G}_1$, either $\hat{\Delta} = r$ or $\hat{\Delta} \leq 4\Delta_{0.8n}(\theta^\star)$. Note that if $\hat{\Delta} \leq 4\Delta_{0.8n}(\theta^\star)$, we can use Equation (26) to provide a multiplicative guarantee. Therefore, conditioned on the event $\mathcal{G}_1$, we have

$$\mathbb{P}\left(F\left(\hat{\theta}; \mathbf{X}^{(n)}\right) - F\left(\theta^\star; \mathbf{X}^{(n)}\right) \leq O\left(\frac{\sqrt{d}}{\sqrt{\rho}}\sqrt{\log(1/\beta)}\right) \cdot r \text{ or}\right.$$

$$\left.F\left(\hat{\theta}; \mathbf{X}^{(n)}\right) \leq \left(1 + O\left(\frac{\sqrt{d}}{n\sqrt{\rho}}\sqrt{\log(1/\beta)}\right)\right)\min_{\theta \in \mathbb{R}^d} F\left(\theta; \mathbf{X}^{(n)}\right) \Big| \mathcal{G}_1\right) \geq 1 - \beta/2. \quad (27)$$

The first statement then follows because, from Theorem 2.7, we have $\mathbb{P}(\mathcal{G}_1) \geq 1 - \beta$.

For the second statement, under the condition that $\max_{i \in [n]} |\mathbf{X}^{(n)} \cap \mathcal{B}_d(x_i, r)| < 3n/4$, we can define the following high probability event:

$$\mathcal{G}_2 = \left\{ \theta^\star \in \Theta_{\text{loc}} \text{ and } \Delta_{0.75n}(\theta^\star) \leq 4\hat{\Delta} \text{ and } \hat{\Delta} \leq 4\Delta_{0.8n}(\theta^\star) \right\}.$$

The argument then proceeds in the same way as the argument for the first claim.

$\square$

*Proof of Lemma 3.2.* We can write $\pi(\cdot; \mathbf{X}^{(n)})$ as

$$\pi(i; \mathbf{X}^{(n)}) = \frac{\exp\left(-\frac{\varepsilon}{2\Delta}\left[F(\theta_i; \mathbf{X}^{(n)}) - F(\theta^\star; \mathbf{X}^{(n)})\right]\right)}{\sum_{j \in [k]} \exp\left(-\frac{\varepsilon}{2\Delta}\left[F(\theta_j; \mathbf{X}^{(n)}) - F(\theta^\star; \mathbf{X}^{(n)})\right]\right)}, \quad \forall i \in [k]. \tag{28}$$

It follows because $F(\theta^\star; \mathbf{X}^{(n)})$ is a constant independent of $i$. Then, the first claim follows from the standard utility analysis of the exponential mechanism in [MT07].

In the next step, we provide the proof for the second claim. To this end, because of Equation (28), we analyze the sensitivity of $F(\theta_i; \mathbf{X}^{(n)}) - F(\theta^\star; \mathbf{X}^{(n)})$ for every $i \in [k]$. Note that $\theta^\star$ is a data dependent quantity. Let $\tilde{\mathbf{X}}^{(n)}$ be a dataset that differ in one sample from $\mathbf{X}^{(n)}$. Also, let $\theta^\circledast \in \text{GM}(\tilde{\mathbf{X}}^{(n)})$ and assume, without loss of generality, that $\tilde{\mathbf{X}}^{(n)} = (x_1, \ldots, x'_n)$ and $\mathbf{X}^{(n)} = (x_1, \ldots, x_n)$. For a fixed $\theta \in \{\theta_1, \ldots, \theta_k\}$, we can write

$$\left[F(\theta; \mathbf{X}^{(n)}) - F(\theta^\star; \mathbf{X}^{(n)})\right] - \left[F(\theta; \tilde{\mathbf{X}}^{(n)}) - F(\theta^\circledast; \tilde{\mathbf{X}}^{(n)})\right]$$

$$= \|\theta - x_n\| - \|\theta - x'_n\| - \|\theta^\star - x_n\| + \|\theta^\circledast - x_{n+1}\| + \sum_{i=1}^{n-1}\left(\|\theta^\circledast - x_i\| - \|\theta^\star - x_i\|\right)$$

$$\leq \|\theta - \theta^\star\| + \|\theta - \theta^\circledast\| + \sum_{i=1}^{n}\left(\|\theta^\circledast - x_i\| - \|\theta^\star - x_i\|\right) \tag{29}$$

$$\leq 2\|\theta - \theta^\star\| + \|\theta^\star - \theta^\circledast\| + \sum_{i=1}^{n}\left(\|\theta^\circledast - x_i\| - \|\theta^\star - x_i\|\right).$$

Here, the last two steps follow from the triangle inequality. Note that $\theta^\circledast$ is the geometric median of $\tilde{\mathbf{X}}^n$. Therefore by the first-order optimally condition, we have

$$\sum_{i=1}^{n-1} \nabla\left(\|\theta^\circledast - x_i\|\right) = -\nabla\left(\|\theta^\circledast - x'_n\|\right). \tag{30}$$

Using the first-order convexity condition applied to the function $h(\theta) = \|\theta - x\|$ for a fixed $x$, we can write

$$\sum_{i=1}^{n-1}\left(\|\theta^\circledast - x_i\| - \|\theta^\star - x_i\|\right) \leq \sum_{i=1}^{n-1}\left\langle \nabla\left(\|\theta^\circledast - x_i\|\right), \theta^\circledast - \theta^\star\right\rangle$$

$$= -\left\langle \nabla\left(\|\theta^\circledast - x'_n\|\right), \theta^\circledast - \theta^\star\right\rangle \tag{31}$$

$$\leq \|\theta^\circledast - \theta^\star\|,$$

where the second step follows from Equation (30) and the last step follows because $\|\nabla\left(\|\theta^\circledast - x_{n+1}\|\right)\| \leq 1$. Therefore, using Equation (29) and Equation (31), we have

$$\left[F(\theta; \mathbf{X}^{(n)}) - F(\theta^\star; \mathbf{X}^{(n)})\right] - \left[F(\theta; \tilde{\mathbf{X}}^{(n)}) - F(\theta^\circledast; \tilde{\mathbf{X}}^{(n)})\right] \leq 2\|\theta - \theta^\star\| + 2\|\theta^\star - \theta^\circledast\|.$$

In the next step of the proof, we invoke Lemma C.4 to upperbound the sensitivity as follows

$$2\|\theta - \theta^\star\| + 2\|\theta^\star - \theta^\circledast\| \leq 2\text{diam} + 3\Delta_{3n/4}(\theta^\star)$$

$$\leq \Delta,$$

where the last step follows because $\|\theta - \theta^\star\| \leq \text{diam}$ by the assumption. Notice that the sensitivity analysis in the reverse direction is also the same. Therefore, the second claim follows from the standard analysis of the privacy of the exponential mechanism. $\square$

*Proof of Theorem 3.3.* The privacy proof of Algorithm 4 is relatively non-standard. Let $\mathcal{A}_1\big(\mathbf{X}^{(n)}\big) = \big(\hat{\Delta}, \{\theta_i\}_{i\in\{0,\ldots,k_{\text{ft}}-1\}}\big)$. Also, let $\mathcal{A}_2\big(\mathbf{X}^{(n)}; \big(\hat{\Delta}, \{\theta_i\}_{i\in\{0,\ldots,k_{\text{ft}}-1\}}\big)\big) = \hat{t}$. In particular, $\mathcal{A}_1\big(\mathbf{X}^{(n)}\big)$ can be viewed as the first part of Algorithm 4 before Line 10. Also, $\mathcal{A}_2(\cdot\,;\cdot)$ denotes the exponential mechanism in Line 10 of Algorithm 4. Using the conversion between zCDP and DP, the privacy budget allocation, and the composition properties of zCDP, we have that $\mathcal{A}_1(\cdot)$ satisfies $(\varepsilon/2, \delta/2)$-DP.

Define the following event

$$\mathcal{G} = \{\theta^\star \in \Theta_0 \text{ and } \Delta_{0.75n}(\theta^\star) \le 4\hat{\Delta}\}.$$

Let $\mu$ be a measure on $\mathcal{M}_1(\mathbb{R} \times (\mathbb{R}^d)^{k_{\text{ft}}})$ that satisfies the following: for every dataset $\mathbf{X}^{(n)}$, $\mathbb{P}\big(\mathcal{A}_1(\mathbf{X}^{(n)}) \in \cdot\big) \ll \mu(\cdot)$. Let $P_1$ denote the density. Since $\mathcal{A}_1$ satisfies approximate-DP, we assume for every $z \in \mathbb{R} \times (\mathbb{R}^d)^{k_{\text{ft}}}$, we have $P_1(z; \mathbf{X}^{(n)}) \le \exp(\varepsilon/2)P_1(z; \tilde{\mathbf{X}}^{(n)}) + \delta/2$.

To prove the requirement of privacy, let $\mathcal{S} \subseteq \mathbb{R} \times (\mathbb{R}^d)^{k_{\text{ft}}} \times \{0, \ldots, k_{\text{ft}} - 1\}$. Also, let $\tilde{\mathbf{X}}^{(n)}$ be a dataset of size $n$ that differs in one sample from $\mathbf{X}^{(n)}$. Then, we can write

$$\mathbb{P}_{\mathcal{A}_1(\mathbf{X}^{(n)}),\mathcal{A}_2(\cdot;\mathbf{X}^{(n)})}\Big(\big(\hat{\Delta}, \{\theta_i\}_{i\in\{0,\ldots,k_{\text{ft}}-1\}}, \hat{t}\big) \in \mathcal{S}\Big)$$

$$= \sum_{\hat{t}} \int \mathbb{1}[\mathcal{S}] P_1\Big(\hat{\Delta}, \{\theta_i\}_{i\in\{0,\ldots,k_{\text{ft}}-1\}} | \mathbf{X}^{(n)}\Big) \cdot \pi\Big(\hat{t} | \hat{\Delta}, \{\theta_i\}_{i\in\{0,\ldots,k_{\text{ft}}-1\}}, \mathbf{X}^{(n)}\Big) d\mu$$

$$= \sum_{\hat{t}} \int \mathbb{1}[\mathcal{S}] P_1\Big(\hat{\Delta}, \{\theta_i\}_{i\in\{0,\ldots,k_{\text{ft}}-1\}} | \mathbf{X}^{(n)}\Big) \cdot \pi\Big(\hat{t} | \hat{\Delta}, \{\theta_i\}_{i\in\{0,\ldots,k_{\text{ft}}-1\}}, \mathbf{X}^{(n)}\Big) \mathbb{1}\Big[(\hat{\Delta}, \theta_0) \in \mathcal{G}\Big] d\mu$$

$$+ \sum_{\hat{t}} \int \mathbb{1}[\mathcal{S}] P_1\Big(\hat{\Delta}, \{\theta_i\}_{i\in\{0,\ldots,k_{\text{ft}}-1\}} | \mathbf{X}^{(n)}\Big) \cdot \pi\Big(\hat{t} | \hat{\Delta}, \{\theta_i\}_{i\in\{0,\ldots,k_{\text{ft}}-1\}}, \mathbf{X}^{(n)}\Big) \mathbb{1}\Big[(\hat{\Delta}, \theta_0) \in \mathcal{G}^c\Big] d\mu$$

$$= \sum_{\hat{t}} \int \mathbb{1}[\mathcal{S}] P_1\Big(\hat{\Delta}, \{\theta_i\}_{i\in\{0,\ldots,k_{\text{ft}}-1\}} | \mathbf{X}^{(n)}\Big) \cdot \pi\Big(\hat{t} | \hat{\Delta}, \{\theta_i\}_{i\in\{0,\ldots,k_{\text{ft}}-1\}}, \mathbf{X}^{(n)}\Big) \mathbb{1}\Big[(\hat{\Delta}, \theta_0) \in \mathcal{G}\Big] d\mu + \mathbb{P}(\mathcal{G}^c).$$

Notice that under the event that $(\hat{\Theta}, \theta_0) \in \mathcal{G}$, we can invoke Lemma 3.2 to reason about the privacy properties of $\mathcal{A}_2$. Under the event $\mathcal{G}$, we can see that $3\Delta_{3n/4}(\theta^\star) + 2\text{diam}(\Theta_0) \le 112\hat{\Delta}$. Therefore, by Lemma 3.2, we have

$$\pi\Big(\hat{t} | \hat{\Delta}, \{\theta_i\}_{i\in\{0,\ldots,k_{\text{ft}}-1\}}, \mathbf{X}^{(n)}\Big) \le \exp(\varepsilon/2) \cdot \pi\Big(\hat{t} | \hat{\Delta}, \{\theta_i\}_{i\in\{0,\ldots,k_{\text{ft}}-1\}}, \tilde{\mathbf{X}}^{(n)}\Big).$$

Moreover, by Theorem 2.7, we have $\mathbb{P}(\mathcal{G}^c) \le \delta/2$. Therefore,

$$\sum_{\hat{t}} \int \mathbb{1}[\mathcal{S}] P_1\Big(\hat{\Delta}, \{\theta_i\}_{i\in\{0,\ldots,k_{\text{ft}}-1\}} | \mathbf{X}^{(n)}\Big) \cdot \pi\Big(\hat{t} | \hat{\Delta}, \{\theta_i\}_{i\in\{0,\ldots,k_{\text{ft}}-1\}}, \mathbf{X}^{(n)}\Big) d\mu$$

$$\le \exp(\varepsilon/2) \sum_{\hat{t}} \int \mathbb{1}[\mathcal{S}] P_1\Big(\hat{\Delta}, \{\theta_i\}_{i\in\{0,\ldots,k_{\text{ft}}-1\}} | \mathbf{X}^{(n)}\Big) \cdot \pi\Big(\hat{t} | \hat{\Delta}, \{\theta_i\}_{i\in\{0,\ldots,k_{\text{ft}}-1\}}, \tilde{\mathbf{X}}^{(n)}\Big) d\mu + \delta/2.$$

Then, we use the fact that $\mathcal{A}_1\big(\mathbf{X}^{(n)}\big)$ satisfies $(\varepsilon/2, \delta/2)$:

$$\exp(\varepsilon/2) \sum_{\hat{t}} \int \mathbb{1}[\mathcal{S}] P_1\Big(\hat{\Delta}, \{\theta_i\}_{i\in\{0,\ldots,k_{\text{ft}}-1\}} | \mathbf{X}^{(n)}\Big) \cdot \pi\Big(\hat{t} | \hat{\Delta}, \{\theta_i\}_{i\in\{0,\ldots,k_{\text{ft}}-1\}}, \tilde{\mathbf{X}}^{(n)}\Big) d\mu + \delta/2$$

$$\le \exp(\varepsilon) \sum_{\hat{t}} \int \mathbb{1}[\mathcal{S}] P_1\Big(\hat{\Delta}, \{\theta_i\}_{i\in\{0,\ldots,k_{\text{ft}}-1\}} | \tilde{\mathbf{X}}^{(n)}\Big) \cdot \pi\Big(\hat{t} | \hat{\Delta}, \{\theta_i\}_{i\in\{0,\ldots,k_{\text{ft}}-1s\}}, \tilde{\mathbf{X}}^{(n)}\Big) d\mu + \delta.$$

It concludes the proof. $\qquad\qquad\qquad\qquad\qquad\qquad\qquad\qquad\qquad\qquad\qquad\qquad\qquad\square$

*Proof of Theorem 3.4.* Define the following event

$$\mathcal{G}_1 = \Big\{\theta^\star \in \Theta_{\text{loc}} \text{ and } \Delta_{0.75n}(\theta^\star) \le 4\hat{\Delta} \text{ and } \Big\{\hat{\Delta} \le 4\Delta_{0.8n}(\theta^\star) \text{ or } \hat{\Delta} = r\Big\}\Big\}. \qquad (32)$$

By Theorem 2.7, and the assumption on the minimum number of samples, we have $\mathbb{P}(\mathcal{G}) \ge 1 - \beta$.

By applying the standard tail bound on the norm of a Gaussian random vector (as outlined in Corollary C.3), we have

$$\mathbb{P}(\mathcal{G}_{n,1}) \triangleq \mathbb{P}\left(\forall t \in \{0, \ldots, k_{\text{ft}} - 1\} : \|\xi_{\text{dir},t}\|^2 \leq \frac{3dk_{\text{ft}}}{2\rho}\left(1 + 4\sqrt{\frac{\log(10k_{\text{ft}}/\beta)}{d}}\right)\right) \tag{33}$$

$$\geq 1 - \beta/5.$$

Also, we can write

$$\mathbb{P}\left(\exists t \in \{0, \ldots, k_{\text{ft}} - 1\} : \langle\xi_{\text{dir},t}, \theta^\star - \theta_t\rangle > 50\hat{\Delta} \cdot \sqrt{\frac{3k_{\text{ft}}}{\rho}\log(10k_{\text{ft}}/\beta)}\right)$$

$$\leq \mathbb{P}\left(\exists t \in \{0, \ldots, k_{\text{ft}} - 1\} : \langle\xi_{\text{dir},t}, \theta^\star - \theta_t\rangle > 50\hat{\Delta} \cdot \sqrt{\frac{3k_{\text{ft}}}{\rho}\log(10k_{\text{ft}}/\beta)}\Big|\mathcal{G}_1\right) + \mathbb{P}(\mathcal{G}_1^c)$$

$$\leq \mathbb{P}\left(\exists t \in \{0, \ldots, k_{\text{ft}} - 1\} : \langle\xi_{\text{dir},t}, \theta^\star - \theta_t\rangle > 50\hat{\Delta} \cdot \sqrt{\frac{3k_{\text{ft}}}{\rho}\log(10k_{\text{ft}}/\beta)}\Big|\mathcal{G}_1\right) + \beta,$$

where the last step follows because $\mathbb{P}(\mathcal{G}_1^c) \leq \beta$. Conditioned on $\mathcal{G}_1$, for all $t \in \{0, \ldots, k_{\text{ft}-1}\}$, we have $\|\theta_t - \theta^\star\| \leq 50\hat{\Delta}$ since $\theta^\star \in \Theta_0$, $\theta_t \in \Theta_0$, and the diameter of $\Theta_0$ is $50\hat{\Delta}$. Also, notice that $\xi_{\text{dir},t} \perp\!\!\!\perp (\theta_t, \Theta_0)$. Using these observations, conditioned on the event $\mathcal{G}_1$, using the standard tail bound on Gaussian random variable (as outlined in Lemma C.1), we can write

$$\mathbb{P}\left(\exists t \in \{0, \ldots, k_{\text{ft}} - 1\} : \langle\xi_{\text{dir},t}, \theta^\star - \theta_t\rangle > 50\hat{\Delta} \cdot \sqrt{\frac{3k_{\text{ft}}}{\rho}\log(10k_{\text{ft}}/\beta)}\Big|\mathcal{G}_1\right) \leq \beta/5.$$

Therefore, we conclude

$$\mathbb{P}(\mathcal{G}_{n,2}) \triangleq \mathbb{P}\left(\forall t \in \{0, \ldots, k_{\text{ft}} - 1\} : \langle\xi_{\text{dir},t}, \theta^\star - \theta_t\rangle \leq 50\hat{\Delta} \cdot \sqrt{\frac{3k_{\text{ft}}}{\rho}\log(10k_{\text{ft}}/\beta)}\right) \tag{34}$$

$$\geq 1 - 6\beta/5.$$

To prove the claim regarding the suboptimality gap, we consider two cases:

1. There exists $t \in \{0, \ldots, k_{\text{ft}} - 1\}$ such that $\theta^\star \in \Theta_t$ and $\theta^\star \notin \Theta_{t+1}$,

2. $\theta^\star \in \Theta_{k_{\text{ft}}}$.

Note that these two events are mutually exclusive and their union covers all the space. In what follows, we show that in both cases there exists $t \in \{0, \ldots, k_{\text{ft}} - 1\}$ such that $F(\theta_t; \mathbf{X}^{(n)})$ has a small excess loss.

For the first case, suppose $t$ be such that $\theta^\star \in \Theta_t$ and $\theta^\star \notin \Theta_{t+1}$. Therefore, we can write

$$\theta^\star \notin \Theta_{t+1} \Leftrightarrow \left\langle\nabla F(\theta_t; \mathbf{X}^{(n)}) + \xi_{\text{dir},t}, \theta^\star - \theta_t\right\rangle \geq 0$$
$$\Leftrightarrow \left\langle\nabla F(\theta_t; \mathbf{X}^{(n)}), \theta^\star - \theta_t\right\rangle \geq -\langle\xi_{\text{dir},t}, \theta^\star - \theta_t\rangle. \tag{35}$$

Notice that using the first-order convexity condition, we have $F(\theta_t; \mathbf{X}^{(n)}) - F(\theta^\star; \mathbf{X}^{(n)}) \leq \left\langle\nabla F(\theta_t; \mathbf{X}^{(n)}), \theta_t - \theta^\star\right\rangle$. Therefore, by Equation (35), we have

$$F(\theta_t; \mathbf{X}^{(n)}) - F(\theta^\star; \mathbf{X}^{(n)}) \leq \left\langle\nabla F(\theta_t; \mathbf{X}^{(n)}), \theta_t - \theta^\star\right\rangle$$
$$\leq \langle\xi_{\text{dir},t}, \theta^\star - \theta_t\rangle. \tag{36}$$

Under the events $\mathcal{G}_1$ and $\mathcal{G}_{n,2}$, defined in Equations (32) and (34), we have

$$F(\theta_t; \mathbf{X}^{(n)}) - F(\theta^\star; \mathbf{X}^{(n)}) \leq \langle\xi_{\text{dir},t}, \theta^\star - \theta_t\rangle \leq \hat{\Delta} \cdot O\left(\sqrt{\frac{k_{\text{ft}}}{\rho}\log(k_{\text{ft}}/\beta)}\right). \tag{37}$$

For the second case, i.e., $\theta^\star \in \Theta_{k_{\text{ft}}}$, we have the following geometric fact [Nes98]: there exists $t \in \{0, \ldots, k_{\text{ft}} - 1\}$ such that the distance of $\theta^\star$ and the separating hyperplane at time $t$ satisfies

$$-\nu \leq \left\langle \frac{\nabla F(\theta_t; \mathbf{X}^{(n)}) + \xi_{\text{dir},t}}{\left\| \nabla F(\theta_t; \mathbf{X}^{(n)}) + \xi_{\text{dir},t} \right\|}, \theta^\star - \theta_t \right\rangle \leq 0. \tag{38}$$

Here $\nu$ is a constant such that $\nu^d \geq \exp(-\tau k_{\text{ft}})(25\hat{\Delta})^d$. The values of $\nu$ and $k_{\text{ft}}$ will be determined later. Using the first order convexity, we can write

$$F(\theta^\star; \mathbf{X}^{(n)}) - F(\theta_t; \mathbf{X}^{(n)})$$

$$\geq \left\| \nabla F(\theta_t; \mathbf{X}^{(n)}) + \xi_{\text{dir},t} \right\| \left\langle \frac{\nabla F(\theta_t; \mathbf{X}^{(n)}) + \xi_{\text{dir},t}}{\left\| \nabla F(\theta_t; \mathbf{X}^{(n)}) + \xi_{\text{dir},t} \right\|}, \theta^\star - \theta_t \right\rangle - \langle \xi_{\text{dir},t}, \theta^\star - \theta_t \rangle$$

$$\geq -\nu \left\| \nabla F(\theta_t; \mathbf{X}^{(n)}) + \xi_{\text{dir},t} \right\| - \langle \xi_{\text{dir},t}, \theta^\star - \theta_t \rangle$$

$$\geq -\nu \left( 2 \left\| \nabla F(\theta_t; \mathbf{X}^{(n)}) \right\| + 2\|\xi_{\text{dir},t}\| \right) - \langle \xi_{\text{dir},t}, \theta^\star - \theta_t \rangle$$

$$\geq -\nu(2n + 2\|\xi_{\text{dir},t}\|) - \langle \xi_{\text{dir},t}, \theta^\star - \theta_t \rangle,$$

where the second step follows from the well-known inequality $\|a + b\| \leq 2\|a\| + 2\|b\|$ for every $a, b \in \mathbb{R}^d$. Then, the last step follows because for every $\theta \in \mathbb{R}^d$, $\left\| \nabla F(\theta; \mathbf{X}^{(n)}) \right\| \leq n$. Therefore, under the events $\mathcal{G}_1$, $\mathcal{G}_{n,1}$, and $\mathcal{G}_{n,2}$, defined in Equations (32) to (34), we have the following bound on the suboptimality gap

$$F(\theta_t; \mathbf{X}^{(n)}) - F(\theta^\star; \mathbf{X}^{(n)}) \leq \nu(2n + 2\|\xi_{\text{dir},t}\|) + \langle \xi_{\text{dir},t}, \theta^\star - \theta_t \rangle$$

$$\leq \nu(2n + 2\|\xi_{\text{dir},t}\|) + O\left( \hat{\Delta} \sqrt{\frac{k_{\text{ft}}}{\rho} \log(k_{\text{ft}}/\beta)} \right)$$

$$\leq \nu \cdot O\left( n + \sqrt{\frac{dk_{\text{ft}}}{\rho} \left( 1 + \sqrt{\frac{\log(k_{\text{ft}}/\beta)}{d}} \right)} \right) + O\left( \hat{\Delta} \sqrt{\frac{k_{\text{ft}}}{\rho} \log(k_{\text{ft}}/\beta)} \right). \tag{39}$$

Recall that $\nu$ satisfies $\nu^d \geq \exp(-\tau k_{\text{ft}})(25\hat{\Delta})^d$. It can be easily seen that by setting

$$k_{\text{ft}} = \Theta\left( \frac{d}{\tau} \log\left( \frac{n\sqrt{\rho}}{\sqrt{d}} + \sqrt{d} \right) \right),$$

under the events $\mathcal{G}_1$, $\mathcal{G}_{n,1}$, and $\mathcal{G}_{n,2}$, we can further upperbound Equation (39) as follows

$$F(\theta_t; \mathbf{X}^{(n)}) - F(\theta^\star; \mathbf{X}^{(n)}) \leq O\left( \hat{\Delta} \sqrt{\frac{k_{\text{ft}}}{\rho} \log(k_{\text{ft}}/\beta)} \right). \tag{40}$$

Therefore, from Equations (37) and (40), under the event $\mathcal{G}_1 \cap \mathcal{G}_{n,1} \cap \mathcal{G}_{n,2}$, for both cases we showed that there exists $t$ such that

$$F(\theta_t; \mathbf{X}^{(n)}) - F(\theta^\star; \mathbf{X}^{(n)}) \leq \Delta_{0.8n}(\theta^\star) \cdot O\left( \sqrt{\frac{k_{\text{ft}}}{\rho} \log(k_{\text{ft}}/\beta)} \right)$$

$$\text{or } F(\theta_t; \mathbf{X}^{(n)}) - F(\theta^\star; \mathbf{X}^{(n)}) \leq r \cdot O\left( \sqrt{\frac{k_{\text{ft}}}{\rho} \log(k_{\text{ft}}/\beta)} \right). \tag{41}$$

By the non-negativity of $\|\cdot\|_2$, we have

$$F(\theta^\star; \mathbf{X}^{(n)}) = \sum_{i=1}^{n} \|\theta^\star - x_i\| \geq 0.2n\Delta_{0.8n}(\theta^\star). \tag{42}$$

Therefore, we conclude that for both cases there exists $t$ such that

$$F(\theta_t; \mathbf{X}^{(n)}) \leq \left( 1 + O\left( \frac{1}{n} \sqrt{\frac{d \log(\kappa)}{\tau\rho}} \cdot \log\left( \frac{d}{\tau\beta} \log(\kappa) \right) \right) \right) F(\theta^\star; \mathbf{X}^{(n)})$$

$$\text{or } F(\theta_t; \mathbf{X}^{(n)}) - F(\theta^\star; \mathbf{X}^{(n)}) \leq r \cdot O\left( \sqrt{\frac{d \log(\kappa)}{\tau\rho}} \cdot \log\left( \frac{d}{\tau\beta} \log(\kappa) \right) \right) \tag{43}$$

where $\kappa \triangleq \frac{n\sqrt{\rho}}{\sqrt{d}} + \sqrt{d}$.

Let us define $\text{OPT} \triangleq \min_{t \in \{0,\ldots,k_{\text{ft}}-1\}} \{F(\theta_t; \mathbf{X}^{(n)}) - F(\theta^\star; \mathbf{X}^{(n)})\}$. In the next step of the proof, we show that the exponential mechanism in Line 10 with high probability can identify an iterate whose suboptimality gap is close to OPT. Using the properties of the exponential mechanism in Line 10 as outlied in Lemma 3.2, we have with probability at least $1 - \beta/3$ over the randomness of the exponential mechanism

$$F(\theta_{\hat{t}}; \mathbf{X}^{(n)}) - F(\theta^\star; \mathbf{X}^{(n)}) \leq \text{OPT} + \frac{448\hat{\Delta}}{\varepsilon}(\log(3k_{\text{ft}}/\beta)). \tag{44}$$

Notice that under the event $\mathcal{G}_1$ and using Equation (42), we have

$$\frac{448\hat{\Delta}}{\varepsilon}\log(3k_{\text{ft}}/\beta) \leq \frac{F(\theta^\star; \mathbf{X}^{(n)})}{n\varepsilon} \cdot O(\log(k_{\text{ft}}/\beta)) \text{ or } \frac{448\hat{\Delta}}{\varepsilon}\log(3k_{\text{ft}}/\beta) \leq \frac{r}{\varepsilon}O(\log(k_{\text{ft}}/\beta)) \tag{45}$$

Moreover, under the event $\mathcal{G}_1 \cap \mathcal{G}_{n,1} \cap \mathcal{G}_{n,2}$, we provided an upperbound on OPT in Equation (43). Combining Equation (43), Equation (44), and Equation (45), proves the first claim.

For the second statement, under the condition that $\max_{i \in [n]} |\mathbf{X}^{(n)} \cap \mathcal{B}_d(x_i, r)| < 3n/4$, we can define the following high probability event:

$$\mathcal{G}_2 = \left\{ \theta^\star \in \Theta_{\text{loc}} \text{ and } \Delta_{0.75n}(\theta^\star) \leq 4\hat{\Delta} \text{ and } \hat{\Delta} \leq 4\Delta_{0.8n}(\theta^\star) \right\}.$$

The argument then proceeds in the same way as the argument for the first claim. $\qquad\square$

# F    Proof of Section 4

*Proof of Lemma 4.2.* For $i \in [n-k]$, we can write
$$\|\theta_k - x_i\| \geq \|\theta_k - \theta_0\| - \|\theta_0 - x_i\|$$
$$= \|\theta_k - \theta_0\| - 2\|\theta_0 - x_i\| + \|\theta_0 - x_i\|.$$

Also for every $j \in [k]$, we have
$$\|\theta_k - y_j\| \geq \|\theta_0 - y_j\| - \|\theta_0 - \theta_k\|.$$

Summing both sides of these inequalities, we obtain

$$\sum_{i=1}^{n-k}\|\theta_k - x_i\| + \sum_{j=1}^{k}\|\theta_k - y_j\|$$

$$\geq (n-2k)\|\theta_0 - \theta_k\| - 2\sum_{i=1}^{n-k}\|\theta_0 - x_i\| + \sum_{i=1}^{n-k}\|\theta_0 - x_i\| + \sum_{j=1}^{k}\|\theta_0 - y_j\|$$

$$(\Leftrightarrow) \sum_{i=1}^{n-k}\|\theta_k - x_i\| + \sum_{j=1}^{k}\|\theta_k - y_j\| - \left(\sum_{i=1}^{n-k}\|\theta_0 - x_i\| + \sum_{j=1}^{k}\|\theta_0 - y_j\|\right)$$

$$\geq (n-2k)\|\theta_0 - \theta_k\| - 2\sum_{i=1}^{n-k}\|\theta_0 - x_i\|$$

Since $\sum_{i=1}^{n-k}\|\theta_k - x_i\| + \sum_{j=1}^{k}\|\theta_k - y_j\| - \left[\sum_{i=1}^{n-k}\|\theta_0 - x_i\| + \sum_{j=1}^{k}\|\theta_0 - y_j\|\right] \leq 0$ by the assumption that $\theta_k = \text{GM}(x_1, \ldots, x_{n-k}, y_1, \ldots, y_k)$, we obtain

$$(n-2k)\|\theta_0 - \theta_k\| - 2\sum_{i=1}^{n-k}\|\theta_0 - x_i\| \leq 0$$

$$\Leftrightarrow \|\theta_0 - \theta_k\| \leq \frac{2}{n-2k} \cdot \sum_{i=1}^{n-k}\|\theta_0 - x_i\|$$

$$\Rightarrow \|\theta_0 - \theta_k\| \leq \frac{2}{n-2k} \cdot \left(\sum_{i=1}^{n-k}\|\theta_0 - x_i\| + \sum_{i=n-k+1}^{n}\|\theta_0 - x_i\|\right)$$

$$\Leftrightarrow \|\theta_0 - \theta_k\| \leq \frac{2}{n-2k} \cdot F(\theta_0; (x_1, \ldots, x_n)).$$

$\square$

*Proof of Theorem 4.1.* We claim that for every neighbouring datasets $\mathbf{X} \in (\mathcal{B}_d(R))^n$ and $\mathbf{X}' \in (\mathcal{B}_d(R))^n$ and for every $y \in \mathcal{B}_d(R)$, we have

$$|\text{len}_r(\mathbf{X}, y) - \text{len}_r(\mathbf{X}', y)| \leq 1.$$

This follows from the fact that for every $\tilde{X}$, we have $\text{d}_{\text{H}}(\mathbf{X}, \tilde{\mathbf{X}}) \leq \text{d}_{\text{H}}(\mathbf{X}', \tilde{\mathbf{X}}) + 1$. Then, the proof of privacy follows from the privacy proof of the exponential mechanism [MT07].

Next, we present the utility proof. Let $k \in \mathbb{N}$ be a constant that determined later. Define the following two sets:

$$A_1 = \{y \in \mathcal{B}_d(R) : \text{len}_r(\mathbf{X}, y) \geq k\}$$
$$A_2 = \{y \in \mathcal{B}_d(R) : \text{len}_r(\mathbf{X}, y) = 0\}$$

Then,

$$\frac{\mathbb{P}_{\hat{\theta} \sim \pi}(\hat{\theta} \in A_1)}{\mathbb{P}_{\hat{\theta} \sim \pi}(\hat{\theta} \in A_2)} = \frac{\int_{y \in A_1} \exp\left(-\frac{\varepsilon}{2} \cdot \text{len}_r(\mathbf{X}, y)\right) dy}{\int_{y \in A_2} \exp\left(-\frac{\varepsilon}{2} \cdot \text{len}_r(\mathbf{X}, y)\right) dy}$$
$$\leq \frac{\exp(-\frac{\varepsilon}{2}k) \int_{y \in A_1} dy}{\int_{y \in A_2} dy}. \tag{46}$$

We can use the following simple facts: $\int_{y \in A_1} dy \leq \int_{y \in \mathcal{B}_d(R)} dy = V_1 R^d$ where $V_1$ is the volume of the ball of radius one in $\mathbb{R}^d$. For $A_2$ notice that, for all $y \in \mathcal{B}_d(\text{GM}(\mathbf{X}), r)$, we have $\text{len}_r(\mathbf{X}, y) = 0$. Thus, $\int_{y \in A_2} dy \geq V_1 r^d$. Putting these two pieces together,

$$\frac{\mathbb{P}_{\hat{\theta} \sim \pi}(\hat{\theta} \in A_1)}{\mathbb{P}_{\hat{\theta} \sim \pi}(\hat{\theta} \in A_2)} \leq \exp\left(-\frac{\varepsilon}{2}k\right)\left(\frac{R}{r}\right)^d \Rightarrow \mathbb{P}_{\hat{\theta} \sim \pi}(\hat{\theta} \in A_1) \leq \exp\left(-\frac{\varepsilon}{2}k\right)\left(\frac{R}{r}\right)^d, \tag{47}$$

where the last step follows from the fact that $\mathbb{P}_{\hat{\theta} \sim \pi}(\hat{\theta} \in A_2) \leq 1$. Therefore, we obtain that for every $\beta \in (0, 1)$ with probability at least $1 - \beta$ we have

$$\text{len}_r(\mathbf{X}, \hat{\theta}) \leq \left\lfloor \frac{2}{\varepsilon}\left(\log\left(\frac{1}{\beta}\right) + d \log\left(\frac{R}{r}\right)\right)\right\rfloor \triangleq k^\star, \tag{48}$$

where $\hat{\theta} \sim \pi$.

Under the above event, let $\hat{\theta} \in \mathcal{B}_d(R)$ be such that $\text{len}_r(\mathbf{X}, \hat{\theta}) \leq k^\star$. This is equivalent to the following: there exists $z \in \mathcal{B}_d(\hat{\theta}, r)$ and $\tilde{\mathbf{X}} \in (\mathbb{R}^d)^n$ such that $z = \text{GM}(\tilde{\mathbf{X}})$ and $\text{d}_{\text{H}}(\mathbf{X}, \tilde{\mathbf{X}}) \leq k^\star$. Using this observation, we can write

$$\left\|\hat{\theta} - \text{GM}(\mathbf{X})\right\| \leq \|z - \text{GM}(\mathbf{X})\| + \left\|\hat{\theta} - z\right\|$$
$$\leq \|z - \text{GM}(\mathbf{X})\| + r \tag{49}$$
$$= \left\|\text{GM}(\tilde{\mathbf{X}}) - \text{GM}(\mathbf{X})\right\| + r.$$

**Suboptimality Gap:** Let $\theta^\star \in \text{GM}(\mathbf{X})$, then

$$F(\hat{\theta}; \mathbf{X}) - F(\theta^\star; \mathbf{X}) = \sum_{i=1}^n \left(\left\|\hat{\theta} - x_i\right\| - \|\theta^\star - x_i\|\right)$$
$$\leq \sum_{i=1}^n (\|z - x_i\| + r - \|\theta^\star - x_i\|)$$
$$= nr + \sum_{i=1}^n \left(\left\|\text{GM}(\tilde{\mathbf{X}}) - x_i\right\| - \|\theta^\star - x_i\|\right).$$

Define $\mathcal{I} \subseteq [n]$ be the indices of the points that $\mathbf{X}$ and $\tilde{\mathbf{X}}$ differs. We know that $|\mathcal{I}| \leq k^\star$. Then, we can write

$$\sum_{i=1}^{n}\left(\left\|\mathrm{GM}(\tilde{\mathbf{X}}) - x_i\right\| - \|\theta^\star - x_i\|\right)$$
$$= \underbrace{\sum_{i \in \mathcal{I}}\left(\left\|\mathrm{GM}(\tilde{\mathbf{X}}) - x_i\right\| - \|\theta^\star - x_i\|\right)}_{A_1} + \underbrace{\sum_{i \in [n]/\mathcal{I}}\left(\left\|\mathrm{GM}(\tilde{\mathbf{X}}) - x_i\right\| - \|\theta^\star - x_i\|\right)}_{A_2}. \quad (50)$$

By triangle inequality, we can write

$$A_1 = \sum_{i \in \mathcal{I}}\left(\left\|\mathrm{GM}(\tilde{\mathbf{X}}) - x_i\right\| - \|\theta^\star - x_i\|\right)$$
$$\leq |\mathcal{I}|\left\|\theta^\star - \mathrm{GM}(\tilde{\mathbf{X}})\right\| \quad (51)$$
$$\leq k^\star \cdot \left\|\theta^\star - \mathrm{GM}(\tilde{\mathbf{X}})\right\|.$$

For $i \in \mathcal{I}$, let $(\tilde{\mathbf{X}})_i = x_i'$ where $(\tilde{\mathbf{X}})_i$ denote the $i$-th data point in $\tilde{\mathbf{X}}$. Since $\mathrm{GM}(\tilde{\mathbf{X}})$ is a geometric median of $\tilde{\mathbf{X}}$, by the first-order optimality condition

$$\nabla_\theta F(\mathrm{GM}(\tilde{\mathbf{X}}); \tilde{\mathbf{X}}) = 0 \Leftrightarrow \sum_{i \in [n]/\mathcal{I}} \nabla_\theta\left(\left\|\mathrm{GM}(\tilde{\mathbf{X}}) - x_i\right\|\right) = -\sum_{i \in \mathcal{I}} \nabla_\theta\left(\left\|\mathrm{GM}(\tilde{\mathbf{X}}) - x_i'\right\|\right). \quad (52)$$

To control $A_2$, notice that $\|\theta - x_i\|$ is a convex function in $\theta$ for every $x_i$. By the first-order convexity condition, for every $\theta_1$ and $\theta_2$, we have $\|\theta_1 - x_i\| - \|\theta_2 - x_i\| \leq \langle \nabla(\|\theta_1 - x_i\|), \theta_1 - \theta_2 \rangle$. Therefore, we can write

$$\sum_{i \in [n]/\mathcal{I}}\left(\left\|\mathrm{GM}(\tilde{\mathbf{X}}) - x_i\right\| - \|\theta^\star - x_i\|\right) \leq \sum_{i \in [n]/\mathcal{I}}\left\langle \nabla\left(\left\|\mathrm{GM}(\tilde{\mathbf{X}}) - x_i\right\|\right), \mathrm{GM}(\tilde{\mathbf{X}}) - \theta^\star \right\rangle. \quad (53)$$

Then, by Equation (52),

$$A_2 = \sum_{i \in [n]/\mathcal{I}}\left\langle \nabla\left(\left\|\mathrm{GM}(\tilde{\mathbf{X}}) - x_i\right\|\right), \mathrm{GM}(\tilde{\mathbf{X}}) - \theta^\star \right\rangle$$
$$= -\sum_{i \in \mathcal{I}}\left\langle \nabla_\theta\left(\left\|\mathrm{GM}(\tilde{\mathbf{X}}) - x_i'\right\|\right), \mathrm{GM}(\tilde{\mathbf{X}}) - \theta^\star \right\rangle.$$

Finally notice that by Equation (3), for every $x_i'$, $\left\|\nabla_\theta\left(\left\|\mathrm{GM}(\tilde{\mathbf{X}}) - x_i'\right\|\right)\right\| \leq 1$. Therefore, by Cauchy–Schwarz inequality

$$A_2 = -\sum_{i \in \mathcal{I}}\left\langle \nabla_\theta\left(\left\|\mathrm{GM}(\tilde{\mathbf{X}}) - x_i'\right\|\right), \mathrm{GM}(\tilde{\mathbf{X}}) - \theta^\star \right\rangle$$
$$\leq |\mathcal{I}|\left\|\mathrm{GM}(\tilde{\mathbf{X}}) - \theta^\star\right\| \quad (54)$$
$$\leq k^\star\left\|\mathrm{GM}(\tilde{\mathbf{X}}) - \theta^\star\right\|.$$

By Equations (51) and (54), we obtain

$$F(\hat{\theta}; \mathbf{X}) - F(\theta^\star; \mathbf{X}) \leq nr + 2k^\star\left\|\mathrm{GM}(\tilde{\mathbf{X}}) - \theta^\star\right\|.$$

Then, we invoke Lemma 4.2 which states that $\left\|\mathrm{GM}(\tilde{\mathbf{X}}) - \mathrm{GM}(\mathbf{X})\right\| \leq \dfrac{2}{n - 2k^\star} \cdot F(\mathrm{GM}(\mathbf{X}); \mathbf{X})$. Putting all the pieces together,

$$F(\hat{\theta}; \mathbf{X}) - F(\theta^\star; \mathbf{X}) \leq nr + 2k^\star\left\|\mathrm{GM}(\tilde{\mathbf{X}}) - \theta^\star\right\|$$
$$\leq nr + \frac{4k^\star}{n - 2k^\star} \cdot F(\theta^\star; \mathbf{X}), \quad (55)$$

as was to be shown.

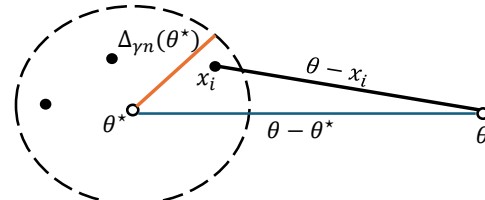

**Figure 1:** Graphical Intuition Behind Equation (57)

**Distance to $\theta^\star$:** From Equation (49), we know that $\left\|\hat\theta - \mathrm{GM}(\mathbf{X})\right\| \leq \left\|\mathrm{GM}(\tilde{\mathbf{X}}) - \mathrm{GM}(\mathbf{X})\right\| + r$ where $\tilde{\mathbf{X}}$ is a dataset of size $n$ such that $\mathrm{d_H}(\mathbf{X}, \tilde{\mathbf{X}}) \leq k^\star$. The proof is based on characterizing the worst case distance between the geometric median of two datasets that differ in at most $k^\star$ points.

For the dataset $\mathbf{X}^{(n)}$, recall that $\theta^\star = \mathrm{GM}\big(\mathbf{X}^{(n)}\big)$. Also, recall the definition of $\Delta_{\gamma n}(\theta^\star)$ from Definition 1.1. Let $\theta \in \mathbb{R}^d$ be such that $\|\theta - \theta^\star\| > \Delta_{\gamma n}(\theta^\star)$. Define $m = |\tilde{\mathbf{X}} \cap \mathcal{B}_d(\theta^\star, \Delta_{\gamma n}(\theta^\star))|$. By the variational representation of $\|\cdot\|$, we can write

$$
\begin{aligned}
\left\|\nabla F(\theta; \tilde{\mathbf{X}})\right\| &\geq \left\langle \nabla F(\theta; \tilde{\mathbf{X}}), \frac{\theta - \theta^\star}{\|\theta - \theta^\star\|} \right\rangle \\
&= \sum_{x \in \tilde{\mathbf{X}} \cap \mathcal{B}_d(\theta^\star, \Delta_{\gamma n}(\theta^\star))} \left\langle \frac{\theta - x}{\|\theta - x\|}, \frac{\theta - \theta^\star}{\|\theta - \theta^\star\|} \right\rangle + \sum_{x \in \tilde{\mathbf{X}} \setminus \{\tilde{\mathbf{X}} \cap \mathcal{B}_d(\theta^\star, \Delta_{\gamma n}(\theta^\star))\}} \left\langle \frac{\theta - x}{\|\theta - x\|}, \frac{\theta - \theta^\star}{\|\theta - \theta^\star\|} \right\rangle \\
&\geq \sum_{x \in \tilde{\mathbf{X}} \cap \mathcal{B}_d(\theta^\star, \Delta_{\gamma n}(\theta^\star))} \left\langle \frac{\theta - x}{\|\theta - x\|}, \frac{\theta - \theta^\star}{\|\theta - \theta^\star\|} \right\rangle - (n - m),
\end{aligned}
$$

(56)

where the last step follows from Cauchy-Schwarz inequality. Then, we claim that for every $x \in \tilde{\mathbf{X}} \cap \mathcal{B}_d(\theta^\star, \Delta_{\gamma n}(\theta^\star))$, we have

$$
\left\langle \frac{\theta - x}{\|\theta - x\|}, \frac{\theta - \theta^\star}{\|\theta - \theta^\star\|} \right\rangle \geq \sqrt{1 - \left(\frac{\Delta_{\gamma n}(\theta^\star)}{\|\theta - \theta^\star\|}\right)^2}
$$

(57)

To gain the intuition behind it see Figure 1. Therefore, from Equation (56),

$$
\left\|\nabla F(\theta; \tilde{\mathbf{X}})\right\| \geq m\sqrt{1 - \left(\frac{\Delta_{\gamma n}(\theta^\star)}{\|\theta - \theta^\star\|}\right)^2} - (n - m).
$$

We are interested on characterizing the condition under which $\left\|\nabla F(\theta; \tilde{\mathbf{X}})\right\| > 0$. A sufficient condition is that given $n < 2m$

$$
m\sqrt{1 - \left(\frac{\Delta_{\gamma n}(\theta^\star)}{\|\theta - \theta^\star\|}\right)^2} - (n - m) > 0 \quad (\Leftrightarrow) \quad \Delta_{\gamma n}(\theta^\star) \frac{1}{\sqrt{2\frac{m}{n} - \left(\frac{m}{n}\right)^2}} < \|\theta - \theta^\star\|.
$$

This shows that the distance of $\mathrm{GM}(\tilde{\mathbf{X}})$ and $\theta^\star$ has to satisfy

$$
\left\|\mathrm{GM}(\tilde{\mathbf{X}}) - \theta^\star\right\| \leq \frac{\Delta_{\gamma n}(\theta^\star)}{\sqrt{2\frac{m}{n} - \left(\frac{m}{n}\right)^2}}.
$$

The function $h(x) = \frac{1}{\sqrt{2x - x^2}}$ is decreasing in the range of $x \in (0, 1]$. Also, notice that $m = |\tilde{\mathbf{X}} \cap \mathcal{B}_d(\theta^\star, \Delta_{\gamma n}(\theta^\star))| \geq \gamma n - k^\star$. Therefore,

$$
\left\|\mathrm{GM}(\tilde{\mathbf{X}}) - \theta^\star\right\| \leq \frac{\Delta_{\gamma n}(\theta^\star)}{\sqrt{2\left(\gamma - \frac{k^\star}{n}\right) - \left(\gamma - \frac{k^\star}{n}\right)^2}},
$$

as was to be shown.

$\square$

# G  Proof of Section 5

*Proof of Theorem 5.1.*  The proof is based on the reduction provided in Lemma G.1 and the lower-bound on the sample complexity of the mean estimation of Gaussian distribution with known covariance matrix in [KLSU19, Thm. 6.5].  □

**Lemma G.1.** *Let $\varepsilon \le 49 \times 10^{-5}$, $\alpha \le 49 \times 10^{-5}$, $\delta \le 10^{-4}$, and $d \ge 22500$ be constants. Let $\mathcal{A}_n$ be an arbitrary $(\varepsilon, \delta)$-DP algorithm such that for every dataset $\mathbf{X}^{(n)}$, its output satisfies*

$$\mathbb{E}_{\hat{\theta} \sim \mathcal{A}_n(\mathbf{X}^{(n)})}\left[ F(\hat{\theta}; \mathbf{X}^{(n)}) \right] \le (1 + \alpha) \min_{\theta \in \mathcal{B}_d^\infty(1)} F(\theta; \mathbf{X}^{(n)}).$$

*Let $\mu \in \mathcal{B}_d^\infty(1)$ and let $\mathbf{X}^{(n)} = (X_1, \ldots, X_n) \sim \mathcal{N}(\mu, \mathbb{I}_d)^{\otimes n}$. Let $\hat{\theta} \sim \mathcal{A}_n(\mathbf{X}^{(n)})$. Then, with probability at least $2/3$ over $\mathbf{X}^{(n)}$ and the internal randomness of $\mathcal{A}_n$, we have*

$$\left\| \hat{\theta} - \mu \right\| \le 0.2\sqrt{d}.$$

*Proof.*  The proof consists of several steps:

**Step 1: Bound on the Empirical Error.**  Let $\mathcal{A}_n$ be an arbitrary $(\varepsilon, \delta)$-DP algorithm such that for every dataset $\mathbf{X}^{(n)}$, its output satisfies

$$F(\hat{\theta}; \mathbf{X}^{(n)}) \le (1 + \alpha) \min_{\theta \in \mathcal{B}_d^\infty(R)} F(\theta; \mathbf{X}^{(n)}).$$

Let $\mu \in \mathcal{B}_d^\infty(R)$ and $\mathbf{X}^{(n)} = (X_1, \ldots, X_n) \sim \mathcal{N}(\mu, \mathbb{I}_d)^{\otimes n}$. The utility guarantee of the algorithm implies that

$$\mathbb{E}\left[ F(\hat{\theta}; \mathbf{X}^{(n)}) \right] \le (1 + \alpha)\mathbb{E}\left[ \min_{\theta \in \mathcal{B}_d^\infty(R)} F(\theta; \mathbf{X}^{(n)}) \right]$$

$$\le (1 + \alpha)\mathbb{E}\left[ F(\mu; \mathbf{X}^{(n)}) \right].$$

To further upperbound the last step, we can use Jensen's inequality to write

$$\frac{1}{n} \cdot \mathbb{E}\left[ F(\mu; \mathbf{X}^{(n)}) \right] = \mathbb{E}[\|X_1 - \mu\|]$$

$$\le \sqrt{\mathbb{E}\left[ \|X_1 - \mu\|^2 \right]}$$

$$= \sqrt{d}$$

Therefore, in-expectation over $\mathbf{X}^{(n)} \sim \mathcal{N}(\mu, \mathbb{I}_d)^{\otimes n}$ and the internal randomness of $\mathcal{A}_n$, we have

$$\mathbb{E}_{\mathbf{X}^{(n)} \sim \mathcal{N}(\mu, \mathbb{I}_d)^{\otimes n}, \hat{\theta} \sim \mathcal{A}_n(\mathbf{X}^{(n)})}\left[ F\left( \hat{\theta}; \mathbf{X}^{(n)} \right) - F\left( \mu; \mathbf{X}^{(n)} \right) \right] \le n\alpha\sqrt{d}. \tag{58}$$

**Step 2:  Relating Empirical Error to Population Error.**  Let $(X_0, X_1, \ldots, X_n) \sim \mathcal{N}(\mu, \mathbb{I}_d)^{\otimes(n+1)}$. With an abuse of notation, let $\theta = \mathcal{A}_n((X_1, \ldots, X_n))$, and, for every $i \in [n]$, let $\theta^{(i)} = \mathcal{A}_n((X_1, \ldots, X_{i-1}, X_0, X_{i+1}, \ldots, X_n))$. Let $T$ be a constant that will be determined later. We can write

$$\mathbb{E}\left[ \left\| \theta^{(i)} - X_i \right\| \right] = \int_{t=0}^\infty \mathbb{P}\left( \left\| \theta^{(i)} - X_i \right\| \ge t \right)\mathrm{d}t$$
$$= \int_{t=0}^T \mathbb{P}\left( \left\| \theta^{(i)} - X_i \right\| \ge t \right)\mathrm{d}t + \int_{t=T}^\infty \mathbb{P}\left( \left\| \theta^{(i)} - X_i \right\| \ge t \right)\mathrm{d}t. \tag{59}$$

Consider the first term in Equation (59). Since $\mathcal{A}_n$ satisfies $(\varepsilon, \delta)$-DP,

$$\mathbb{P}\left( \left\| \theta^{(i)} - X_i \right\| \ge t \right) = \mathbb{E}\left[ \mathbb{P}\left( \left\| \theta^{(i)} - X_i \right\| \ge t \Big| (X_0, \ldots, X_n) \right) \right]$$
$$\le \mathbb{E}\left[ \exp(\varepsilon)\mathbb{P}\left( \|\theta - X_i\| \ge t \Big| (X_0, \ldots, X_n) \right) + \delta \right] \tag{60}$$
$$= \exp(\varepsilon) \cdot \mathbb{P}(\|\theta - X_i\| \ge t) + \delta.$$

Therefore, the first term can be upperbounded as

$$\int_{t=0}^{T} \mathbb{P}\left(\left\|\theta^{(i)} - X_i\right\| \geq t\right) \mathrm{d}t \leq \exp(\varepsilon) \cdot \int_{t=0}^{T} \mathbb{P}(\|\theta - X_i\| \geq t) \mathrm{d}t + T\delta$$
$$\leq \exp(\varepsilon) \cdot \mathbb{E}[\|\theta - X_i\|] + T\delta.$$

In the next step, we upperbound the the second term in Equation (59). Notice that

$$\left\{(\theta^{(i)}, X_i) : \left\|\theta^{(i)} - \mu - (X_i - \mu)\right\| \geq t\right\} \subseteq \left\{(\theta^{(i)}, X_i) : \|X_i - \mu\| \geq t - \left\|\theta^{(i)} - \mu\right\|\right\}$$
$$\subseteq \left\{X_i : \|X_i - \mu\| \geq t - 2R\sqrt{d}\right\}, \tag{61}$$

where the first step follows from the triangle inequality and the last step follows because $\mu$ and $\theta^{(i)}$ are in $\mathcal{B}_d^\infty(R)$. Using this, we can write

$$\int_{t=T}^{\infty} \mathbb{P}\left(\left\|\theta^{(i)} - X_i\right\| \geq t\right) \mathrm{d}t \leq \int_{t=T}^{\infty} \mathbb{P}\left(\|X_i - \mu\| \geq t - 2R\sqrt{d}\right) \mathrm{d}t$$
$$= \int_{u=T-(2R+1)\sqrt{d}}^{\infty} \mathbb{P}\left(\|X_i - \mu\| \geq u + \sqrt{d}\right) \mathrm{d}u, \tag{62}$$

where the last step follows from the change of variable $u = t - (2R+1)\sqrt{d}$. In the next step, we use the concentration bounds for the norm of multivariate Guassian random variable. Using Lemma C.2, we can write

$$\mathbb{P}\left(\|X_i - \mu\| \geq u + \sqrt{d}\right) = \mathbb{P}\left(\|X_i - \mu\|^2 \geq u^2 + d + 2u\sqrt{d}\right)$$
$$\leq \exp\left(-\frac{u^2}{2}\right). \tag{63}$$

Let $T = 2(2R+1)\sqrt{d}$. Then, using standard bounds on the *complementary error function* [Ksc17], we can write

$$\int_{t=T}^{\infty} \mathbb{P}\left(\|X_i - \mu\| \geq u + \sqrt{d}\right) \leq \int_{u=(2R+1)\sqrt{d}}^{\infty} \exp\left(-\frac{u^2}{2}\right) \mathrm{d}u$$
$$\leq \frac{1}{(4R+2)\sqrt{d}} \exp\left(-2(2R+1)^2 d\right). \tag{64}$$

In the last step, we claim that $\mathbb{E}[\|\theta - X_0\|] = \mathbb{E}\left[\left\|\theta^{(i)} - X_i\right\|\right]$ for every $i \in [n]$. It is because $\theta^{(i)} \stackrel{d}{=} \theta$, $X_i \stackrel{d}{=} X_0$, and $\theta^{(i)} \perp\!\!\!\perp X_i$. Ergo, combining and summing over $i \in [n]$, we obtain

$$\mathbb{E}[\|\theta - X_0\|]$$
$$\leq \exp(\varepsilon)\left(\frac{1}{n}\sum_{i=1}^{n} \mathbb{E}[\|\theta - X_i\|]\right) + (4R+2)\sqrt{d}\delta + \frac{1}{(4R+2)\sqrt{d}}\exp\left(-2(2R+1)^2 d\right)$$

This bound implies that

$$\mathbb{E}[\|\theta - X_0\|] - \mathbb{E}[\|\mu - X_0\|]$$
$$\leq \exp(\varepsilon)\left(\frac{1}{n}\sum_{i=1}^{n}(\mathbb{E}[\|\theta - X_i\|] - \mathbb{E}[\|\mu - X_0\|])\right) + (\exp(\varepsilon) - 1)\mathbb{E}[\|\mu - X_0\|]$$
$$+ (4R+2)\sqrt{d}\delta + \frac{1}{(4R+2)\sqrt{d}}\exp\left(-2(2R+1)^2 d\right).$$

This equation can be rephrased as follows

$$\mathbb{E}[\|\theta - X_0\|] - \mathbb{E}[\|\mu - X_0\|] \leq \beta\sqrt{d} \tag{65}$$

where

$$\beta = \exp(\varepsilon)\alpha + (\exp(\varepsilon) - 1) + (4R+2)\delta + \frac{1}{(4R+2)d}\exp\left(-2(2R+1)^2 d\right). \tag{66}$$

**Step 3: Relating Population Error to Distance**   In Step 2, we showed that in-expectation over $(X_0, \ldots, X_n) \sim \mathcal{N}(\mu, \mathbb{I}_d)^{\otimes (n+1)}$ and $\hat{\theta} \sim \mathcal{A}_n(\mathbf{X}^{(n)})$ where $\mathbf{X}^{(n)} = (X_1, \ldots, X_n)$, we have

$$\mathbb{E}\left[\left\|\hat{\theta} - X_0\right\|\right] - \mathbb{E}[\|\mu - X_0\|] \leq \beta\sqrt{d}. \tag{67}$$

For notiational convenience, let $h : \mathbb{R}^d \to \mathbb{R}$ be $h(\theta) \triangleq \mathbb{E}_{X \sim \mathcal{N}(\mu, \mathbb{I}_d)}[\|\theta - X\|]$. Equation (67) can be written as

$$\mathbb{E}_{\mathbf{X}^{(n)} \sim \mathcal{N}(\mu, \mathbb{I}_d)^{\otimes n}, \hat{\theta} \sim \mathcal{A}_n(\mathbf{X}^{(n)})}\left[h(\hat{\theta}) - h(\mu)\right] \leq \beta\sqrt{d}.$$

Since $\mu$ is the minimizer of $h(\theta)$, for every $\theta \in \mathbb{R}^d$ we have that $h(\theta) \geq h(\mu)$. Therefore, $h(\hat{\theta}) - h(\mu)$ is a non-negative random variable. We can invoke Markov's inequality to write

$$\mathbb{P}_{\mathbf{X}^{(n)} \sim \mathcal{N}(\mu, \mathbb{I}_d)^{\otimes n}, \hat{\theta} \sim \mathcal{A}_n(\mathbf{X}^{(n)})}\left(h(\hat{\theta}) - h(\mu) \leq 3\beta\sqrt{d}\right) \geq \frac{2}{3}. \tag{68}$$

In the next step, we provide a *deterministic* argument: For every $\theta \in \mathbb{R}^d$ such that $h(\theta) - h(\mu) \leq 3\beta\sqrt{d}$, we provide an upperbound on $\|\theta - \mu\|$. By subtracting $\mu$, we can write

$$\begin{aligned}
h(\theta) - h(\mu) &= \mathbb{E}_{X \sim \mathcal{N}(\mu, \mathbb{I}_d)}[\|\theta - X\| - \|\mu - X\|] \\
&= \mathbb{E}_{Z \sim \mathcal{N}(0, \mathbb{I}_d)}[\|\theta - \mu + Z\| - \|Z\|].
\end{aligned} \tag{69}$$

Define the following events

$$\begin{aligned}
\mathcal{E}_1 &\triangleq \left\{ Z : \sqrt{d\left(1 - 2\sqrt{\frac{\log(4/\gamma)}{d}}\right)} \leq \|Z\| \leq \sqrt{d\left(1 + 4\sqrt{\frac{\log(4/\gamma)}{d}}\right)} \right\}, \\
\mathcal{E}_2 &\triangleq \left\{ Z : \langle \theta - \mu, Z \rangle \geq -\|\theta - \mu\|\sqrt{2\log(2/\gamma)} \right\}.
\end{aligned} \tag{70}$$

Using Corollary C.3 and simple concentration bound for Gaussian random variable we have that $\mathbb{P}(\mathcal{E}_1 \cap \mathcal{E}_2) \geq 1 - \gamma$. Let $\mathcal{E} = \mathcal{E}_1 \cap \mathcal{E}_2$. By dropping the positive term, we can write

$$\begin{aligned}
&\mathbb{E}[\|\theta - \mu + Z\| - \|Z\|] \\
&= \mathbb{E}[(\|\theta - \mu + Z\| - \|Z\|) \cdot \mathbb{1}[\mathcal{E}]] + \mathbb{E}[(\|\theta - \mu + Z\| - \|Z\|) \cdot \mathbb{1}[\mathcal{E}^c]] \\
&\geq \mathbb{E}[(\|\theta - \mu + Z\| - \|Z\|) \cdot \mathbb{1}[\mathcal{E}]] - \mathbb{E}[\|Z\| \cdot \mathbb{1}[\mathcal{E}^c]].
\end{aligned} \tag{71}$$

Using Cauchy-Schwarz inequality, $\mathbb{E}[\|Z\| \cdot \mathbb{1}[\mathcal{E}^c]] \leq \sqrt{\mathbb{P}(\mathcal{E}^c)}\sqrt{\mathbb{E}[\|Z\|^2]} = \sqrt{\mathbb{P}(\mathcal{E}^c)}\sqrt{d} \leq \sqrt{\gamma}\sqrt{d}$. In the next step, we analyze the first term.

$$\begin{aligned}
&\mathbb{E}[(\|\theta - \mu + Z\| - \|Z\|) \cdot \mathbb{1}[\mathcal{E}]] \\
&= \mathbb{E}\left[\left(\sqrt{\|\theta - \mu\|^2 + \|Z\|^2 + 2\langle \theta - \mu, Z \rangle} - \|Z\|\right) \cdot \mathbb{1}[\mathcal{E}]\right] \\
&= \mathbb{E}\left[\|Z\|\left(\sqrt{1 + \frac{\|\theta - \mu\|^2}{\|Z\|^2} + 2\frac{\langle \theta - \mu, Z \rangle}{\|Z\|^2}} - 1\right) \cdot \mathbb{1}[\mathcal{E}]\right]
\end{aligned} \tag{72}$$

The value of $\gamma$ will be determined later. Let $d$ be large enough such that

$$\left(1 - 2\sqrt{\frac{\log(4/\gamma)}{d}}\right) = 0.9 \quad \text{and} \quad \left(1 + 4\sqrt{\frac{\log(4/\gamma)}{d}}\right) = 1.1. \tag{73}$$

Then, we can write

$$\begin{aligned}
&\mathbb{E}\left[\|Z\|\left(\sqrt{1 + \frac{\|\theta - \mu\|^2}{\|Z\|^2} + 2\frac{\langle \theta - \mu, Z \rangle}{\|Z\|^2}} - 1\right) \cdot \mathbb{1}[\mathcal{E}]\right] \\
&\geq \sqrt{0.9d}\left(\sqrt{1 + \frac{\|\theta - \mu\|^2}{1.1d} - \frac{2\sqrt{2\log(2/\gamma)}\|\theta - \mu\|}{0.9d}} - 1\right).
\end{aligned} \tag{74}$$

Notice that we assumed that $h(\theta) - h(\mu) \leq 3\beta\sqrt{d}$. Therefore, we have

$$\sqrt{0.9d}\left(\sqrt{1 + \frac{\|\theta - \mu\|^2}{1.1d} - \frac{2\sqrt{2\log(2/\gamma)}\|\theta - \mu\|}{0.9d}} - 1\right) - \sqrt{\gamma}\sqrt{d} \leq 3\beta\sqrt{d}$$

$$(\Leftrightarrow)\sqrt{1 + \frac{\|\theta - \mu\|^2}{1.1d} - \frac{2\sqrt{2\log(2/\gamma)}\|\theta - \mu\|}{0.9d}} \leq \frac{(3\beta + \sqrt{\gamma})}{\sqrt{0.9}} + 1. \tag{75}$$

Simple calculations show that this bound implies that

$$\|\theta - \mu\| \leq 3.45\sqrt{\log(2/\gamma)} + \sqrt{1.1d}\sqrt{\left(\left(1 + \frac{3\beta + \sqrt{\gamma}}{\sqrt{0.9}}\right)^2 - 1\right)}$$

$$\leq 3.45\sqrt{\log(2/\gamma)} + 0.1\sqrt{d} \tag{76}$$

$$\leq 15 + 0.1\sqrt{d}.$$

We would like to set the parameters such that $\sqrt{1.1d}\sqrt{\left(\left(1 + \frac{3\beta + \sqrt{\gamma}}{\sqrt{0.9}}\right)^2 - 1\right)} = 0.1\sqrt{d}$. We can easily see that it implies $3\beta + \sqrt{\gamma} = 0.0045$. For example, we can pick $\beta = 0.0014$ and $\gamma = 9 \times 10^{-8}$. Recall that

$$\beta = \exp(\varepsilon)\alpha + (\exp(\varepsilon) - 1) + (4R + 2)\delta + \frac{1}{(4R + 2)d}\exp\left(-2(2R + 1)^2d\right). \tag{77}$$

For instance, by setting $\varepsilon \leq 49 \times 10^{-5}$, $\alpha \leq 49 \times 10^{-5}$, $\delta \leq \frac{2}{3} \times 10^{-4}$ and $d \geq 2$, we obtain $\beta \leq 0.1$. Finally, we need to set $d$ such that Equation (73) holds. We can see that $d \geq 7050$ satisfies this condition. □

## H   Details of the Numerical Experiment

Our goal in the experiments is to evaluate the impact of increasing the radius of the initial feasible set, i.e. $R$, on the performance of our proposed algorithm and compare it with DPGD. Also, we want to show that our method without any additional hyperparmeter tuning can achieve a good excess error.

**Data Generation.**   Let $n$ denote the number of samples. We assume that $0.9n$ of the data is distributed as follows: let $\mu \in \mathbb{R}^d$ be a uniformly random vector within $\mathcal{S}_{d-1}(50)$. We then sample $0.9n$ of the data points from $\mathcal{N}(\mu, (0.01)^2 \cdot \mathbb{I}_d)$. The remaining $0.1n$ of the points are sampled uniformly at random from $\mathcal{B}_d(100)$.

**Hyperparameters.**   We set the discretization parameter to $r = 0.05$ in Algorithm 2 and failure probability to $5\%$. Additionally, we repeat each algorithm 10 times and report the mean. For the other hyperparameters, we used exactly the same hyperparameters as stated in Algorithm 3. For DPGD, we use the hyperparameters in Lemma A.1, and in particular, we choose $T$ such that $\frac{\sqrt{2}}{\sqrt{T}} = \frac{16\sqrt{d}}{n\sqrt{\rho}}$.

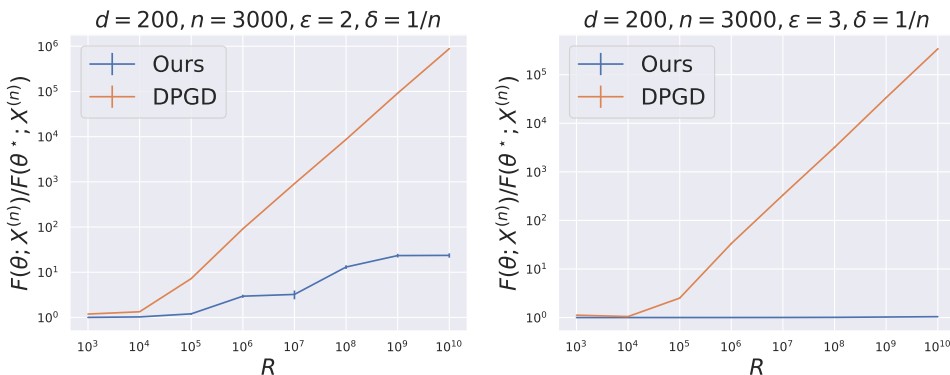

**Figure 2:** Performance for Different Privacy Budget

