# OpenReview forum: "Private Geometric Median"
_NeurIPS.cc/2024/Conference — NeurIPS 2024 poster_

### Official Review · Reviewer_wFAP · 2024-06-25

**Soundness:** 3
**Presentation:** 3
**Contribution:** 3
**Rating:** 7
**Confidence:** 3

**Summary:**

The paper proposes algorithms that compute the geometric median with differential privacy. Composed of two parts, the algorithm firstly finds the quantile radius that covers enough points. Secondly, it fine-tunes the geometric median. The authors suggest two methods for the second part: LocDPGD and LocDP cutting plane methods.

Additionally, the paper shows a pure DP algorithm for computing the geometric median and a lower bound on the sample complexity required to compute the geometric median with differential privacy.

**Strengths:**

This paper studies a well-motivated problem. As mentioned in the paper, a line of works has studied computing private versions of the medians, but little is known about the world when the dimension is larger than one.

This problem is difficult because the runtime of applying vanilla DP-GD is highly affected by outliers. This paper uses two steps: first, it estimates the quantile radius and an initial point that is not greatly affected by outliers, and then it fine-tunes the computation of the result. This idea is intuitive but requires some good techniques. For example, the way they implement the DP cutting plane method by adding Gaussian noise to the gradient that "cuts" the set seems novel and interesting to me.

This paper is also very complete in the sense that it shows results in pure DP setting and lower bound in sample complexity.

From the writing perspective, this paper is well-organized and straightforward to read.

**Weaknesses:**

It is a bit difficult to understand the dependency between parameters in Thm 3.1 and Thm 3.4. It would be very helpful if the authors could make Thm 2.7, 3.1, and 3.4 more readable and easier to compare.

**Questions:**

I am wondering if the authors can explain a bit more about the comparison between LocDPGD and LocDP cutting-plane. For example, you could compare the time complexity and explain under what circumstances one is better than the other.

I am also wondering what part is the bottleneck if someone wants to improve this to a linear run-time.

---

> ### Author Rebuttal · Authors · 2024-08-07
>
> We thank the reviewer for their positive and thorough review. Based on your suggestions, we revised the statements in Thm. 2.7, Thm. 3.1, and Thm. 3.4 to improve readability.
>
>
> > **W1: dependency between parameters in Thm 3.1 and Thm 3.4.**
>
> Note that there is **no** dependency between the parameters in Thm. 3.1 and Thm. 3.4. in the sense that the statements of the theorems are self-contained. Theorems 3.1 and 3.4 concern the utility of the algorithm using DP-(S)GD and DP cutting plane method, respectively. Both of the algorithms have the same utility guarantee. There is an additional log-factor (only in $n,d,\varepsilon$) in the utility guarantee of the cutting plane method.
> Due to the space limitation, in the submitted version of the paper we needed to introduce $\kappa$ and $\alpha$ so that we can fit the equations in the line. In the revised version of the paper, we will improve the readability of the mentioned theorems.
>
> > **Q1:  On comparison of LocDPGD and LocDP.**
>
> Both of the algorithms have the same utility guarantee (up to a log factor only in $n,d,\varepsilon$). However, the runtime is different. Consider the case that $\varepsilon=\Theta(1)$. Then, if we approximate the matrix-multiplication exponent by $2.5$, we have the following:
> Given  $n \leq \tilde{O}(d^{1.75})$, LocDPSGD achieves better runtime, while for $n > \tilde{\Omega}(d^{1.75})$,  localized DP cutting plane methods achieve a better runtime. This comparison is based on the runtimes presented in Table 1. We added this discussion into the paper.
>
> > **Q2:  Bottlenecks for improving runtime.**
>
> There are two bottlenecks in our algorithms:
>
> 1- The first roadblock is that the estimation of quantile radius requires $n^2$ computations as discussed in Remark 2.1. One possible solution here is to consider subsampling for estimating $N_i(v)$, i.e., the number of data points within distance of $v$ from $x_i$. In particular, one can sample a constant number of points and only use this subset to estimate $N_i(v)$. While this approach provides an accurate estimate of $N_i(v)$ with high probability in a non-private setting, it introduces technical challenges for privacy analysis.
>
> 2- The second roadblock is that for private convex optimization of non-smooth loss functions there is **no** general first-order method with an optimal excess error that requires only $n$ gradient evaluations. (see [FKT20] and [CJJLLDT23] for recent developments.) To address this, we need to develop a specific first-order algorithm for the geometric median problem that requires $n$ gradient evaluation to achieve the optimal error.
>
> By addressing the aforementioned challenges, we can develop a linear-time algorithm that achieves optimal excess error for the geometric median problem. We will incorporate this discussion into the revised version of the paper.
>
> [FKT20] Feldman V, Koren T, Talwar K. Private stochastic convex optimization: optimal rates in linear time. InProceedings of the 52nd Annual ACM SIGACT Symposium on Theory of Computing 2020 Jun 22 (pp. 439-449).
>
> [CJJLLDT23] Carmon Y, Jambulapati A, Jin Y, Lee YT, Liu D, Sidford A, Tian K. Resqueing parallel and private stochastic convex optimization. In2023 IEEE 64th Annual Symposium on Foundations of Computer Science (FOCS) 2023 Nov 6 (pp. 2031-2058). IEEE.

---

> > ### Comment · Reviewer_wFAP · 2024-08-10
> >
> > Thank you for your response! Nice work!

---

> > > ### Author Response · Authors · 2024-08-12
> > >
> > > Thank you for your reply! We will incorporate all the suggested revisions into the final version of the paper.

---

### Official Review · Reviewer_wp5E · 2024-07-10

**Soundness:** 4
**Presentation:** 3
**Contribution:** 4
**Rating:** 8
**Confidence:** 4

**Summary:**

This paper introduces a pair of polynomial-time DP algorithms for computing the geometric median of a dataset. The excess error guarantees of the algorithm scale with the effective radius.

The algorithm includes two parts. The first part shrinks the feasible set to a ball whose diameter is proportional to the quantile radius of the optima, which is defined as the radius of the smallest ball containing sufficiently many points. First, it gives a private estimation of the quantile radius. Then, it locates a good initializer based on the estimated quantile radius. AboveThreshold and DP-GD are implemented in the procedure to ensure the approximated DP.

The second part fine-tunes the output of the first part. The paper provides two fine-tuning methods. The first one utilizes DP-GD. The second one modifies the cutting plane method. It adds Gaussian noise to achieve DP. Using a novel sensitivity analysis of the loss function, the authors show that the noise can be scaled down.

This paper also provides the lower bound to demonstrate the optimality of the sample complexity of their algorithm.

**Strengths:**

1. Quality of result:
This paper studies the DP convex optimization task. DP-GD method is well-established in this field. The authors point out that its excess error depends linearly on the radius R of the feasible set. Previous work shows the strongly convex assumption can remove this deficiency, but it is too strong and unrealistic.
This paper improves the dependence on the radius for the geometric median task under weak and natural assumptions. The main contribution of this work is a pair of polynomial-time DP algorithms (with different run-time) for geometric median estimation with an excess error linearly depending on the effective radius r of the dataset, instead of the worst-case radius R. This significantly improves the error bound when r << R. This condition is common since the dataset may have an enormous R due to a small number of outliers.
Other contributions include a pure DP algorithm based on the inverse smooth sensitivity mechanism (yet it is computationally inefficient), and the demonstration of the sample lower bound, which verifies the optimality of the algorithm.

2. Novel techniques: The algorithm's good error bound is based on a structural result stated by the paper: Given a point \theta, if its distance to the optima \theta^* is larger than the quantile radius \Delta_{3n/4}(\theta^*}, it has a linear growth in its loss value: F\left(\theta ; \mathbf{X}^{(n)}\right)-F\left(\theta^{\star} ; \mathbf{X}^{(n)}\right) \gtrsim\left|\theta-\theta^{\star}\right|. This simulates the effect of strongly-convexity, which has a quadratic growth. This implies that one can take a larger step size when performing the DP-GD. It will move fast towards a sub-optimal point. To utilize this property, the authors build a private estimation algorithm to find this radius quantile. Then, the algorithm fine-tunes the sub-optimal point to find a good estimation of the geometric mean. This paper states a modified DP-SGD algorithm to do this task. To reach a better run-time under certain conditions, this paper also provides a private cutting plane method by adding local noise.

3. Technically soundness: This paper is sound technical with solid proof and analysis. It has an informative technical overview. It also includes a numerical experiment to strengthen their result. It compares their algorithm with DPGD in solving the geometric mean task. Under large R, their algorithm shows significant improvement upon DPGD.

4. Reproducibility: The quantile radius technique in the warm-up algorithm is of independent interest. It may be utilized to solve other DP convex optimization tasks. The authors also state the open problem of developing a linear time algorithm with the optimal excess error.

**Weaknesses:**

More words can be added in Section 1.1 to introduce the intuition of their solution (e.g., how to solve the challenge in line 122?). A comparison of run-time and excess risk with methods from previous work may also be helpful.

**Questions:**

N/A

---

> ### Author Rebuttal · Authors · 2024-08-07
>
> We thank the reviewer for their positive and thorough review! We respond to the questions below.
>
> > **W1: Presentation of the ideas behind the algorithms**
>
> Thanks for the suggestion. In the revised version of the paper, we expanded on these challenges and also presented the high level ideas of the proposed solutions. For instance, we added this sentence regarding the challenges discussed in Line 122:
>
> The general analysis of the exponential mechanism involves analyzing the sensitivity of the numerator and normalizing factor separately. We provide a novel analysis of the exponential mechanism with the geometric median loss function as the quality function by **coupling** the sensitivity analysis of the numerator and denominator. In particular, we show that given a set of candidate points that lie in a ball with radius $\Delta$, the noise due to the private selection is only $O(\Delta/\epsilon)$ instead of $O(R/\epsilon)$ one would obtain by the direct application of the exponential mechanism.
>
> > **W2: Comparison with prior work**
>
> Since the geometric median loss function is convex and 1-Lipschitz, one might use an off-the-shelve method for private convex optimization for this task. However, a key shortcoming of this approach is that the excess error scales linearly with $R$. In particular, one can use DP-SGD from [BST14]. The runtime of this method is $n^2 d$ and the excess error is
> $$
> F(\text{DP-SGD};\mathbf{X}^{(n)}) - \min_{\theta \in \mathbb{R}^d} F(\theta;\mathbf{X}^{(n)}) = O\left(\frac{R \sqrt{d}}{\epsilon}\right)
> $$
> Based on your suggestion, we have added a row to Table 1 summarizing the performance of the off-the-shelve method for private convex optimization for this task.

---

> > ### Comment · Reviewer_wp5E · 2024-08-10
> >
> > Thank you for your response! My questions have been addressed.

---

> > > ### Author Response · Authors · 2024-08-12
> > >
> > > Thank you for the reply! We will update the paper based on your suggestions.

---

### Official Review · Reviewer_7SSt · 2024-07-14

**Soundness:** 3
**Presentation:** 3
**Contribution:** 3
**Rating:** 7
**Confidence:** 2

**Summary:**

This paper considers the differentially private geometric median problem. The goal of the geometric median problem is given n data points, find a point to minimize the sum of Euclidean distances from this point to all data points.

The previous DP-GD algorithm requires prior knowledge of the radius R of the data set and also has the excess risk linearly depending on this radius R.

This paper provides three algorithms for the differentially private geometric median. They provide LocDPSGD and LocCuttingPlane algorithms which achieve approximate DP and the excess risk bound depends on the optimal cost instead of the radius R. Finally, they provide an exponential time algorithm that achieves pure DP.

**Strengths:**

1 This paper considers the differentially private geometric median problem, which is an important problem since the geometric median is widely used in many algorithms and requires privacy for many real-world applications.

2 This work provides interesting techniques and algorithms to achieve the excess risk bound depending on the optimal cost instead of the radius of data points R. They developed a private estimation of quantile radius and private localization for the warm-up stage such that the excess risk does not depend on the radius R.

**Weaknesses:**

-

**Questions:**

-

**Limitations:**

Yes.

---

> ### Author Rebuttal · Authors · 2024-08-07
>
> We thank the reviewer for all the positive comments!

---

### Official Review · Reviewer_M9WN · 2024-07-15

**Soundness:** 3
**Presentation:** 2
**Contribution:** 3
**Rating:** 6
**Confidence:** 4

**Summary:**

This paper studies DP algorithms for geometric median. The paper starts with DP-SGD on geometric median as the baseline. The error scales linearly in R, the radius of a ball containing all data points. This can be a very loose bound since the geometric mean is known to be robust to outliers and one single outlier can make this radius R to be arbitrarily large.

The algorithms proposed in this paper has both multiplicative error and additive error. The idea is to replace the estimate of R by the quantile radius -- the value \Delta such that a ball of radius \Delta has at least a certain fraction of the points. This comes with a cost of a multiplicative error.

The algorithm has two phases, with the first phase to estimate the quantile radius in a private manner and the second phase to find the geometric median. The first phase uses techniques from [NSV16]. For the second phase the paper reports several ideas: 1) just apply DP-SGD using the quantile radius; 2) uses a cutting plane method (non-gradient descent method) and 3) for pure DP uses inverse smooth sensitivity of [AD20] -- this one is not efficient in terms of running time.

In summary I think this paper did an OK job but the ideas are in general incremental. I would give a positive rating but I am not extremely enthusiastic (or lacks excitement).

**Strengths:**

The paper has merits of developing DP algorithms for geometric medians. The algorithms have both multiplicative error and additive errors. Adding these algorithms to the repository of DP algorithms for this problem is nice.

**Weaknesses:**

The idea of using "robust" notion of radius, essentially the quantile radius, is not new and has been a standard in many prior work on clustering algorithms -- search clustering with outliers. It will be good to include citation and discussion on that.

The writing can be improved. There are many places where clarification and definition will be helpful to readers. Some examples are mentioned below.

On DP-SGD, may you explain what is the definition of "excess error" (line 23)? Is this the same as excess risk (line 45)?

Equation (2), what is A_n? the DP -SGD algorithm? It is never defined.

Line 63, effective diameter of the data points -- it will be good to explain what effective diameter means.

Table 1, define lower case r? Again r does not appear until much later in the paper.

**Questions:**

See above.

---

> ### Author Rebuttal · Authors · 2024-08-07
>
> We would like to thank the reviewer for their thorough review and valuable feedback. Below, we address each of the points raised.
>
> > **Q1: Robust Notion of Radius and Our Contribution**
>
> We agree with the reviewer that proposing a new robust notion of radius is not a contribution of this work. Our primary contribution lies in connecting the notion of quantile radius to the convergence of gradient-based methods. We establish several fundamental properties of the quantile radius and leverage these properties to design private algorithms:
>
> For instance, 1) the geometric median function satisfies a **growth condition** for a point $\theta$ such that $\lVert \theta - \theta^\star \rVert \gtrsim \Delta_{\gamma n}(\theta^\star)$. (See Lemma  2.6 for a formal statemtn ).This is a key result for developing the warm-up algorithm. It lets us show that we can design gradient-based methods that use large steps sizes with the goal of consuming less privacy budget.  2) Quantile radius is a data-dependent quantity but one can privately estimate it using pairwise distances between the data points. In Lemma 2.3, we show that computing these pairwise distances serves as an effective proxy for determining the quantile radius, which again relies on specific properties of quantile radius.
>
> In summary, we emphasize that the main contribution of our paper is not introducing a new notion of radius. Instead, we demonstrate that the quantile radius has several interesting connections to various properties of the geometric median function, and we exploit these connections to develop a pair of polynomial-time and sample-efficient private algorithms.
>
> Based on your suggestion, we have added a discussion on related work in clustering in the non-private case. In particular, we will mention in the revised version of the paper that the robust notion of radius appears in clustering with outliers algorithms as well (for instance, [BHI02]). We have already discussed the work on the private version of clustering, such as [NSV16; NS18; CKMST21; TCKMS22]. We appreciate the reviewer’s suggestion to include further work on robust clustering.
>
> [BHI02] Bādoiu M, Har-Peled S, Indyk P. Approximate clustering via core-sets. InProceedings of the thiry-fourth annual ACM symposium on Theory of computing 2002 May 19 (pp. 250-257).
>
>
> > **Q2: Excess Error and Excess Risk**
>
> Thank you for catching this typo. We use the notion of excess error of Algorithm $\mathcal{A}$ to refer the achievable cost function compared to the global optimum, i.e., $ F(\mathcal{A}_n;\mathbf{X}^{(n)}) -  F(\theta^\star;\mathbf{X}^{(n)}) $ . We fixed it.
>
> > **Q3: $\mathcal{A}_n$ in Equation 2**
>
> Thank you for pointing this out. As you mentioned it is DP-SGD. We fixed this in the paper.
>
> > **Q4: Effective Diameter**
>
> In Line 63, effective diameter of the data points is an informal way to refer to the quantile radius. Later in Line 77, we mentioned that quantile radius formalizes the idea of effective diameter. Based on your comment, we added a sentence to help a reader understand better the effective diameter.
>
> > **Q5: $r$ in Table 1**
>
> $r$ can be thought of as a very small constant that shows the minimum separation of the data points. More precisely, assume that no data point has 3n/4 of other data points within a ball of radius $r$ centered on it. Then, as we show in the second part of Theorem 3.1., our algorithm has a “purely” multiplicative error. However, in order to get a completely **general result** without placing any assumption on the dataset, we need to incur an additive error proportional to $r$. Note that as the sample complexity and runtime only depends on $\log(1/r)$, we can assume it is very small. Introducing $r$ is unavoidable due to the impossibility results in [NSV16].
>
> Based on your suggestion to improve readability, we have removed $r$ from the summary of the results in Table 1 in the introduction and only present the multiplicative error guarantee. We defer the general results, including the additive error, to the later sections.

---

> > ### Comment · Reviewer_M9WN · 2024-08-12
> >
> > Thanks for the response. Please include these revision items in the paper.

---

> > > ### Author Response · Authors · 2024-08-12
> > >
> > > Thank you for your helpful feedback! We will include a discussion on the robust notions of radius and update the introduction based on your suggestions.

---

### Official Review · Reviewer_3ZzL · 2024-07-16

**Soundness:** 3
**Presentation:** 3
**Contribution:** 3
**Rating:** 7
**Confidence:** 3

**Summary:**

This paper studies differentially private algorithms for computing geometric median (GM) of a dataset.  Previous methods such as DP-SGD requires knowing in advance that all data points live in a ball of radius R (contribution bounding), and the resulting utility guarantee also has a dependency on that R, which can get very big even though the majority of the points are concentrated around the GM. The paper proposes three algorithms, two for approximate DP and one for pure DP. For approximate DP, we run a pre-step for finding a quantile radius which covers the majority of the points and then find a good initialization point using this radius. From there we can either run DP-SGD or using an DP-adapted cutting plane algorithm. For pure-DP the authors describe a discretized inverse-sensitivity mechanism.

**Strengths:**

The problem solved in this paper is a very interesting problem, that is important in practice, and the solution provided in this paper provides the first performance guarantee without the dependency on the spread of dataset that can be achieved with an implementable, polynomial-time algorithm, which is impressive.

While developing the algorithms the paper made many observations that are insightful and could be worth knowing for the community. I think the way it relates the quantile radius around the GM to the average of top m quantile radii of all points in the dataset is clever, and so is the relationship between distance from GM and median cost expressed in terms of quantile radius.

Although there are certain complicated aspects of algorithms, the overall quality of writing is good.

**Weaknesses:**

I found the pure-DP algorithm not as interesting as the approximate one for a couple of reasons: (1) it requires an oracle finding exact GM, which is not known, so I'm wondering what happens if we are given an approximation but then the definition of $len_r(X, y)$ is messed up; (2) the paper does not talk about how to compute $len_r(X, y)$; which I think is non-trivial? (3) is the purpose of introducing $r$ mostly to discretize the original inverse sensitivity mechanism? what is the benefit of using $r$ compared to $r \to 0$?

Numerical setting is relatively simple but I don't really mind that since the greatest merit of this paper is in the theory.

**Questions:**

It seems that the authors have used $\rho-zCDP$ and $(\epsilon, \delta)$-DP interchangeably but without providing preliminaries on the connection between the two? It's better to unify.

For others, see the weakness part.

**Limitations:**

Yes.

---

> ### Author Rebuttal · Authors · 2024-08-07
>
> We thank the reviewer for their positive and thorough review. We respond to the main points below.
>
> > **On Our Pure-DP Algorithm**
>
> Our pure DP algorithm is not the main focus of our paper. Our main result is a pair of polynomial time and sample-efficient algorithms for this problem using gradient methods under the constraint of approximate-DP. However, it is important to understand the achievable utility-privacy tradeoff under the stringent notion of pure-DP.  Our pure-DP algorithm is not computationally efficient, but it shows that under pure-DP, we can achieve a similar data-dependent utility guarantee with $\sqrt(d)$ more samples. The analysis of this algorithm relies on the several robustness properties of the geometric median, which might be of independent interest.
>
> > **W1: Oracle access to exact GM**
>
> Thank you for raising this point! We can relax the assumption of having access to an exact geometric median oracle. Instead, consider an oracle such that for every data set $\mathbf{X}^{(n)}$, outputs $\theta$ such that $\lVert \theta  - \mathrm{GM}(\mathbf{X}^{(n)}) \rVert \leq \alpha$, i.e., the output has distance $\alpha$ from the exact geometric median. We can modify the definition of the length function to use such an oracle. Obviously, inexact oracle changes the utility guarantees and we get an additional additive error. We did not develop this relaxation as it would add complexity without providing additional insight.
>
> > **W2: Computing $\text{len}_r$ function**
>
> Computing this function is not efficient in general. This is a well-known limitation of the inverse sensitivity approach. Indeed, most of the DP algorithms that try to adapt to local sensitivity, such as smooth sensitivity [NRS07] and propose-test-release [DR09], have a similar step that one needs to compute such a length function.
>
> [NRS07] Nissim K, Raskhodnikova S, Smith A. Smooth sensitivity and sampling in private data analysis. InProceedings of the thirty-ninth annual ACM symposium on Theory of computing 2007 Jun 11 (pp. 75-84).
>
> [DR09] Dwork C, Lei J. Differential privacy and robust statistics. InProceedings of the forty-first annual ACM symposium on Theory of computing 2009 May 31 (pp. 371-380).
>
> > **W3: Benefit of r>0**
>
> Introducing $r>0$ is necessary for analyzing the algorithm: In the analysis of the algorithm based on sampling, one can come up with some pathological datasets such that the normalization factor for the sampling distribution becomes infinity. A standard technique to address this is to smooth out the cost function using $r>0$. Notice that the sample complexity of this algorithm scales with $log(1/r)$ as shown in Theorem 4.1. Therefore, we need to assume $r$ is bounded away from zero.
>
>
>
>
> > **Q1: Privacy definitions**
>
> Thanks for the suggestion! We already included in Appendix 3 (A.3) definitions of approximate-DP and zCDP as well as the connection between these two notions. Based on your suggestion, in the final version of this paper, we will add this appendix to the main body.

---

> > ### Comment · Reviewer_3ZzL · 2024-08-08
> > **Thank you for your response**
> >
> > I've read the authors' response and my questions have been addressed adequately. I will keep my current score at 7. Thank you!

---

> > > ### Author Response · Authors · 2024-08-12
> > >
> > > Thank you very much for your reply. We will revise our paper according to the constructive comments in your reviews.

---

### Decision · Program_Chairs · 2024-09-25

**Decision:**

Accept (poster)

**Comment:**

All reviewers agree that the goal of improving the accuracy for the private geometric median problem with a large number of inliers is a well-motivated problem. This paper presents algorithms with various trade-offs and excess error guarantees that scale with the quantile radius, i.e., the effective diameter of the input dataset. The initial reviewer concerns were successfully addressed by the authors during the following discussion.

I would encourage the authors to address reviewer concerns and make improvements to the overall presentation, to strengthen the potential impact of the work.